

# Rainfall Estimates on a Gridded Network (REGEN) - A global land-based gridded dataset of daily precipitation from 1950–2013

Steefan Contractor[1,2], Markus G. Donat[1,3,4], Lisa V. Alexander[1,3], Markus Ziese[5], Anja Meyer-Christoffer[5], Udo Schneider[5], Elke Rustemeier[5], Andreas Becker[5], Imke Durre[6], and Russell S. Vose[6]

[1]Climate Change Research Centre, UNSW Sydney, Australia
[2]ARC Centre of Excellence for Climate System Science
[3]ARC Centre of Excellence for Climate Extremes
[4]Barcelona Supercomputing Center, Barcelona, Spain
[5]Global Precipitation Climatology Centre, Deutscher Wetterdienst, Offenbach Germany
[6]National Centers for Environmental Information, National Oceanic and Atmospheric Administration, Asheville NC, USA

**Correspondence:** Steefan Contractor (s.contractor@unsw.edu.au)

**Abstract.** We present a new global land-based daily precipitation dataset from 1950 using an interpolated network of *in situ* data called **R**ainfall **E**stimates on a **G**ridd**E**d **N**etwork - **REGEN**. We merged multiple archives of *in situ* data including two of the largest archives, the Global Historical Climatology Network - Daily (GHCN-Daily) hosted by National Centres of Environmental Information (NCEI), USA and one hosted by the Global Precipitation Climatology Centre (GPCC) operated by

5     Deutscher Wetterdienst (DWD). This resulted in an unprecedented station density compared to existing datasets. The station timeseries were quality controlled using strict criteria and flagged values were removed. Remaining values were interpolated to create area average estimates of daily precipitation for global land areas on a $1° \times 1°$ latitude-longitude resolution. Besides the daily precipitation amounts, fields of standard deviation, Kriging error and number of stations are also provided. We also provide a quality mask based on these uncertainty measures. For those interested in a dataset with lower station network variability

10     we also provide a related dataset based on a network of long-term stations which interpolates stations with a record length of at least 40 years. The REGEN datasets are expected to contribute to the advancement of hydrological science and practice by facilitating studies aiming to understand changes and variability in several aspects of daily precipitation distributions, extremes, and measures of hydrological intensity. Here we document the development of the dataset and guidelines for best practices for users with regards to the two datasets.

15     *Copyright statement.*

# 1 Introduction

Earth's climate is changing leading to spatial and temporal variations in precipitation. These changes in precipitation are strongly linked to social, economic and environmental prosperity due to the role precipitation plays in global food production



and maintaining biodiversity. Theoretical expectations are that the global hydrological cycle would intensify in a warmer climate, associated with increases in mean and extreme precipitation (whereby mean/total precipitation would increase at lower rate than extreme precipitation due to energetic constraints (Allen and Ingram, 2002)). In addition to changes in precipitation due to climate change, precipitation is also characterised by strong variability in many regions. Reliable observations are

necessary to understand these short- and long-term changes and to evaluate climate models which help understand the processes driving these changes. Hence in some ways gridded observations of the past also help us to better plan for and adapt to these changes in the future.

There are alternative sources to gauge-based precipitation observations such as satellite observations, model reanalysis products and radar-based observations, however they all have limitations. Reanalysis products assimilate observations and models

to create a synthesised estimate of the state of the earth system. They are often considered as the "true state of the system" but in fact inherit issues from the incomplete observations and imperfect models and are based on complex assimilation techniques. Furthermore, none of the reanalysis products assimilate surface precipitation observations and as such are not representative of reality. This is evidenced by the classification of precipitation as the least reliable class by Kalnay et al. (1996). Renalyses also contain temporal inhomogeneities due to the changing amount of assimilated observations over time (Compo et al., 2006).

According to Lorenz and Kunstmann (2012) even the state-of-the-art reanalyses are unsuitable for climate trend and long-term water budget analysis. Radar based observations provide highly accurate and high spatial and temporal resolution estimates for rainfall over local regions but very few national networks of Radar observations exist, with temporal coverage just exceeding 20 years, and no long-term global radar-based datasets exist.

Satellite products have become available in recent years. These datasets are gridded and boast a global if not quasi-global

coverage. The Tropical Rainfall Measuring Mission (TRMM) 3B42 (Huffman et al., 2007), Global Precipitation Climatology Projects 1 Degree Daily (GPCP-1DD) (Huffman et al., 2001), Climate Hazards Group InfraRed Precipitation with Stations (CHIRPS) (Funk et al., 2015) and the Precipitation Estimates from Remotely Sensed Information using Artificial Neural Networks - Climate Data Record (PERSIANN-CDR) (Ashouri et al., 2014) are some examples of popular satellite based precipitation products. These satellite based datasets, however, use complex algorithms to derive precipitation estimates from

indirect radiation measurements resulting in large uncertainties in precipitation estimates. For example GPCP-1DD measures infrared reflectivity of clouds to infer the cloud thickness and then estimates precipitation rates based on the poor relationship between clouds and rainfall (Kidd and Levizzani, 2011). This estimate is also adjusted based on monthly gauge observations, however, the uncertainties remain high. In general satellite products perform well in the tropics where the rain rates are higher but struggle with snow and ice and on complex terrain (Bytheway and Kummerow, 2013; Tian and Peters-Lidard, 2010; Con-

tractor et al., 2015). The biggest limitation of satellite products, however, is also their brevity. It was only after Tropical Rainfall Measurement Mission (TRMM) in 1997 where we entered an era of multi-sensor measurements across multiple satellites to produce a globally consistent and complete map of precipitation (Tian and Peters-Lidard, 2010). Thus the satellite products do not allow for an analysis of global rainfall changes that effectively separates the natural variability from anthropogenic climate change.



Observations have shown spatially varying changes in mean precipitation across the globe (Trenberth, 2011; Hartmann et al., 2013) and robust increases in extreme precipitation across various regions and in the global average (Groisman et al., 2005; Westra et al., 2013; Donat et al., 2016). These global analyses of observed precipitation changes were based on datasets of monthly precipitation accumulations (such as Climate Research Unit's CRU TS (Harris et al., 2014; Mitchell and Jones, 2005),

Global Precipitation Climatology Centre's GPCC Full Data Monthly (Becker et al., 2013; Schneider et al., 2015), Global Historical Climatology Network's GHCN-Monthly (Peterson and Vose, 1997), Global Precipitation Climatology Project GPCP-Monthly (Adler et al., 2003; Huffman et al., 1997) and the Smith et al. (2012) dataset), or datasets providing indices representing specific aspects of extreme precipitation (such as GHCNDEX (Donat et al., 2013a), HadEX (Alexander et al., 2006) and HadEX2 (Donat et al., 2013b)). Availability of daily precipitation data, however, would allow analysis of precipitation at

different parts of the distribution, and for a wider range of temporal aggregations. A daily resolution dataset would also enable a more robust estimate of the extremes since monthly datasets average out the extremes and dampen the variability in daily observations. Existing gauge-based quasi-global gridded datasets of daily precipitation are short (such as CPC Global Precipitation dating back to 1979 (Chen and Xie, 2008; Xie et al., 2007; Chen et al., 2008) and GPCC Full Data Daily V1 which dates back to 1988 (Schamm et al., 2015). An updated version, GPCC Full Data Daily V2018, was released in June 2018,

covering now from 1982 to 2016) and therefore do not allow for robust analysis of long-term variability or trends. The main reason for this is the lack of data sharing between countries which results in poor spatial coverage earlier in time. Even in cases where meteorological organisations have agreements in place with countries to obtain gauge data (such as GPCC on behalf of Deutscher Wetterdienst - DWD), the length of their analysis is limited due to the lack of high quality data extending back in time. To reach a high level of quality, the GPCC applies a quality control procedure with manual inspection of questionable

values, which is very time consuming but preserves the real extremes in the data. Many regional or continental scale products are also available which are produced by local meteorological organisations or researchers who have a more complete set of daily gauge data available to them and thus have longer temporal records. Examples of such datasets include E-OBS for Europe (Haylock et al., 2008), CPC for United States (Chen and Xie, 2008; Xie et al., 2007; Chen et al., 2008), AWAP for Australia (Jones et al., 2009), APHRODITE for Asia (Yatagai et al., 2012), CLARIS for South America (Menendez et al., 2010), as well

as national and regional products for UK (Perry and Hollis, 2005), Spain (Herrera et al., 2012), Germany (Rauthe et al., 2013), Switzerland (Frei and Schär, 1998; Isotta et al., 2013), Norway (Lussana et al., 2018), India (Rajeevan et al., 2006) and Middle East (Yatagai et al., 2008).

Spatially regular gridded data, rather than irregular station data, facilitate many studies (such as climate variability studies investigating connections between regional or global precipitation phenomena and large scale changes) that are not spatially

biased. Furthermore, climate models also rely on gridded data. Gridded datasets are needed for initialising, forcing and validating global and regional climate models. Since the models also produce outputs representative of area averages (Osborn and Hulme, 1998) as opposed to point based processes, gridded datasets are also necessary to evaluate them. Finally, gridded observations can provide a reasonable estimates in regions where local station data is unavailable but stations within the typical length scales of precipitation systems in that region may be present.





Given all the limitations of existing datasets noted above, our aim here was to create a new long-term global land-based dataset with increased raw station density back to the mid-twentieth century. In this study we present the data and methods used to create such a dataset called Rainfall Estimates on a Gridded Network (REGEN) and evaluate it against existing daily and monthly, global and regional products. We also describe how uncertainty estimates are calculated and finally provide

guidelines for how to best use (and not use) the dataset.

## 2   Data and Methods

REGEN was created by acquiring daily station precipitation data from various sources, quality controlling them using an automated algorithm and merging them into a single archive, which was then interpolated with ordinary block Kriging. We created two related datasets, the first dataset interpolated the entire station network referred to henceforth as REGEN and the

second dataset interpolated only the long-term stations referred to henceforth as REGEN40YR. Stations considered long-term here are those with at least 40 complete years of data, described in more detail in 2.4. In this section the various raw data sources, automated quality control, automated station matching algorithm and the interpolation method are described.

### 2.1   Raw Gauge Data

The raw station data for REGEN has 3 sources:

1. the Global Historical Climatology Network - Daily (GHCN-Daily) version 3.22-upd-2017092104: stations hosted by National Centers for Environmental Information (NCEI) in USA (Menne et al., 2012).

2. the Global Precipitation Climatology Centre (GPCC), operated by Deutscher Wetterdienst (DWD) and

3. Other

The total number of stations interpolated each day in REGEN range from a minimum of 35,460 to a maximum of 56,190,

with an average of 50,530 (figure 1a). Regionally, the number of stations per day doubles in North America after 2000 and decreases substantially in South America from the late 1990s. There are no Chinese stations in 1950 and there is a large drop in stations in India in 1970 affecting the total number of stations per day in Asia. Stations in Africa are sparse throughout the time period of REGEN, however there are still more stations compared to other existing global rainfall products. However, this highlights a very important issue regarding the sharing of meteorological data between countries. Global datasets of

observations are limited by the amount of station data available. Regions of poor station coverage are most abundant in Africa and Asia because of limited capability or readiness of countries to share data, despite the World Meteorological Organisation (WMO) data policy encouraging free and unrestricted exchange of meteorological data and products. Therefore, even the *in situ* data held by GPCC can only be distributed in the form of derived products such as the gridded dataset described in this article.

The majority of the underlying station data for REGEN is sourced from the stations hosted by GPCC (figure 1b). Note that figure 1b does not show the actual number of stations in GHCN-Daily or Other archives, but rather the number of daily



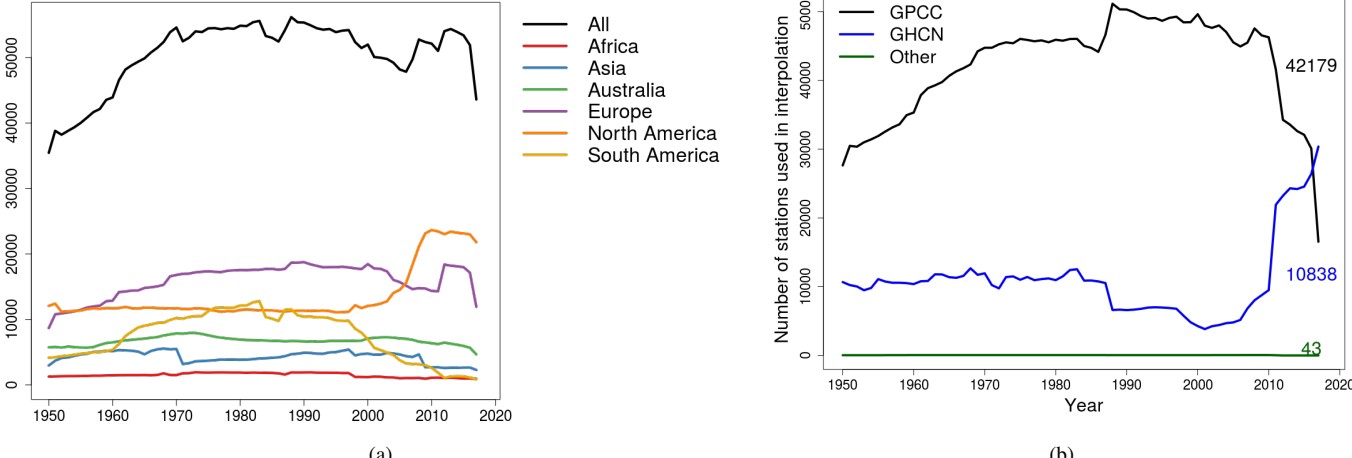

**Figure 1.** Final (interpolated), quality-controlled number of stations over time by (a) region and (b) source. Figure 4c shows a map of the regions. Due to the varying station network over time, the total number of stations over the entire temporal domain sums to 135,178 stations. The numbers in black, blue and green in (b) refer to the average number of stations from GPCC, GHCN and Other sources respectively.

records from stations in GHCN-Daily or Other that were unique with respect to the stations in the GPCC archive. Due to the large overlap between the archives, the number of stations from GHCN-Daily is higher when fewer stations from GPCC are available. There is a gradual increase in stations from GPCC until 1990 and a steep decline after 2010. All quality controlled station data hosted by GPCC are eventually archived in a relational database (henceforth referred to as GPCC data base), however, there were additional ASCII data files for various countries that were not processed at the time of the analysis (henceforth referred to as GPCC ASCII data files).

Figure 2 shows that most of the station data in Central America, western South America, Europe, Africa, Middle East and East Asia was sourced from GPCC.

We summarise the spatial and temporal distribution of the station network comprising REGEN in figure 3. Each map in figure 3 refers to a decade and shows for each grid the percentage of days in each decade with at least one station, based on REGEN (figure 3a), REGEN40YR (figure 3b) and also GPCC's Full Data Daily V1 (GPCC-FDD1; (Schamm et al., 2015)) for comparison (figure 3c). We compare REGEN's station network with GPCC-FDD1's because until REGEN, GPCC-FDD1 was the global dataset of daily precipitation with the highest station density. It can be seen that not only is REGEN's station network density higher than GPCC-FDD1 in all the decades, but even the REGEN40YR station network with a much stricter completeness criteriion has more stations in all three comparable decades relative to GPCC-FDD1.

## 2.2 Quality Control

The quality control procedures used in REGEN were adopted from NCEI, part of National Oceanic and Atmospheric Administration (NOAA) in USA (Durre et al., 2010). The quality control is done in two stages and climatologies generated in an





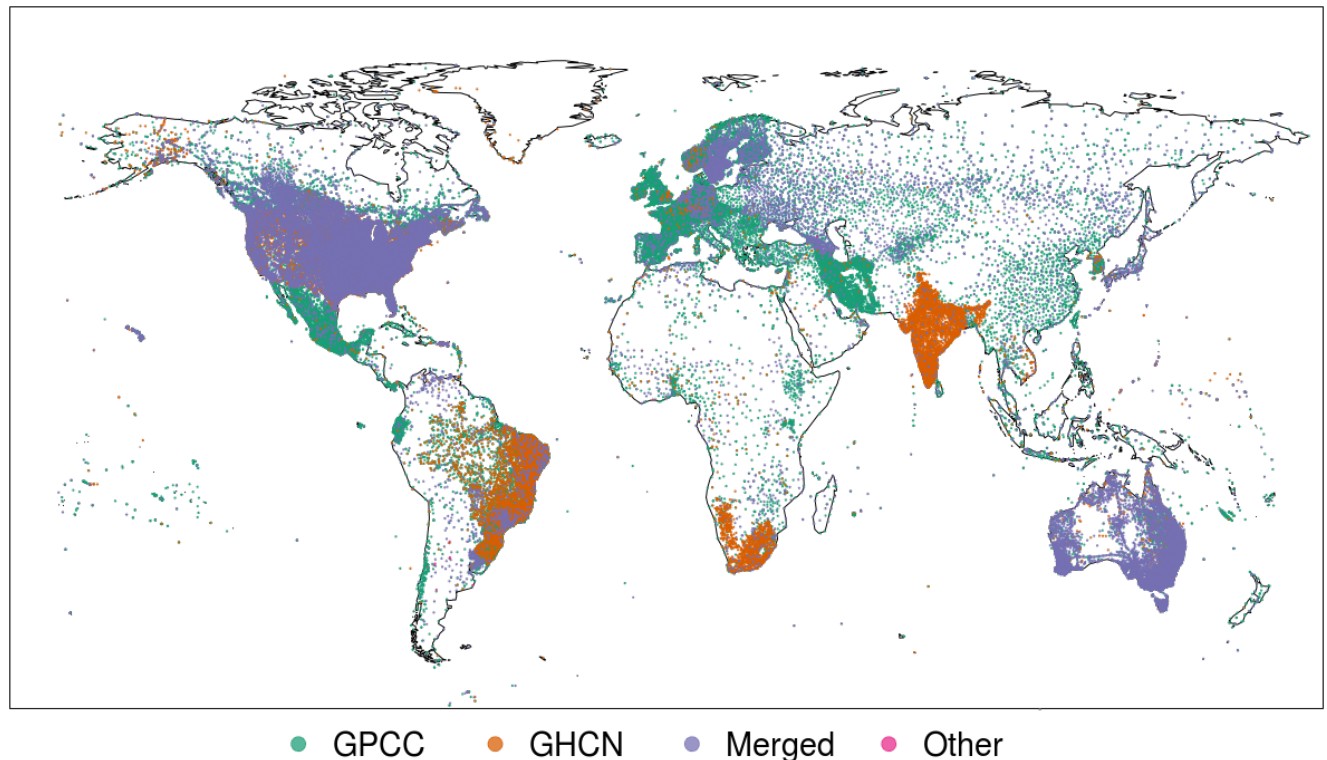

**Figure 2.** Distribution of stations color coded by source. "GPCC" refers to stations hosted by Deutsche Wetterdienst, "GHCN" refers to stations hosted by National Centers for Environmental Information (NCEI), "Merged" refers to stations that have been identified as identical in two or more archives resulting in a merger of the timeseries and finally "Other" refers to the Russian and Argentinian stations that were added by us.

auxiliary step are used in both stages. At the end of the quality control process all data are written in a common format identical to the GHCN-Daily format (see README file, ftp://ftp.ncdc.noaa.gov/pub/data/ghcn/daily/readme.txt).

The first quality control stage involves basic integrity checks such as checks for erroneous zeros, conflicts between multi-day accumulations and daily reports, duplication of entire years or months, repetition or frequent occurrence of values, and world record exceedances. In addition this test stage also checks for outliers by checking for gaps in tails of distributions and checks for climatological outliers. The test also performs some temporal consistency checks by comparing values with consecutive days to look for unrealistic spikes in precipitation. The second quality control stage does spatial corroboration checks which determines if the value at each station is consistent with the values at neighbouring stations. For further information and detail on the quality control algorithms, refer to Durre et al. (2010). Data failing any tests at any point of the quality control process are flagged (see GHCN-Daily README file (ftp://ftp.ncdc.noaa.gov/pub/data/ghcn/daily/readme.txt) for a list of quality flags and their meanings). In order to ensure a high quality final dataset, all flagged data are removed prior to interpolation. Although the QC procedures were designed to minimize the number of instances in which true extremes are flagged as errors (Durre



et al., 2010), it is possible that a few such extremes are among the flagged values that were withheld from the REGEN input data. Future versions of REGEN may consider methods for recognising and saving possible flagged extremes.

All data sources (each country in the GPCC ASCII data, the GPCC data base and "Other" data) were quality controlled individually before merging. Since our QC prodcedures are identical to the GHCN-Daily, we used the flags already included

with the GHCN-Daily data. The percentage of flagged records per year in the final merged input data average around 0.05-0.06% throughout the time period spiking to 0.1% around 2010 (figure 4a). This may be because the number of stations in the final merged station network sourced from GHCN-Daily increase in time in the last decade of the temporal record while the number of stations sourced from GPCC decrease. Since GPCC data are assumed to be of higher quality compared to GHCN-Daily due to the manual quality control it is subjected to, the flag rate increases with time as well due to the higher percentage

of GHCN-Daily stations in the last decade of the final merged station network. In general we also see a trend of increasing missing months with time in all regions (figure 4b). A month is marked as missing if it contains fewer than 70% of the possible number of daily data records. As a result the percentage of missing months is also an indicator of the completeness of the daily data records. The spike in missing month percentage in South Asia is due to the drop in station data from India in the 1970s.

### 2.3    Merger of GHCN-Daily, GPCC and other smaller data archives

Once the station data from various sources were quality controlled individually they were merged with each other in multiple steps. First, the manually and automatically quality controlled data in GPCC's data base were merged with ASCII data files for various countries that at the time of the analysis were not integrated into the GPCC data base, to create a combined archive of quality controlled GPCC stations. This GPCC archive was then merged with GHCN-Daily archive and subsequently the Argentinian and Russian data respectively.

For consistent comparison GPCC shifts data for certain countries so the daily amount always represents the day closest to 7am the day of the timestamp to 7am the next day, local time. For example, if the source *in situ* data timestamp represents the day from 9am the previous day to 9am the day of the source timestamp, then the resulting GPCC timestamps are shifted a day back compared to the source timestamps. This results in climatologically consistent timestamps. In our case while merging the GHCN data, we shifted the GHCN data timestamps identically to the way GPCC shifted their timestamps, for all countries

whose timestamps were shifted by GPCC. The countries for which the data are shifted a day back (e.g. data from 2$^{nd}$ Jan are saved as 1$^{st}$ Jan) are listed in the Appendix. So far, the data from no country are shifted forward. This data shifting is important to keep in mind when comparing REGEN with regional datasets. For example when comparing REGEN with the precipitation from the Australian Water Availability Project regional dataset (AWAP; (Jones et al., 2009)) we shifted AWAP a day backward. This may also result in inconsistent comparisons between REGEN and satellite datasets which represent

UTC0 the day of the timestamp to UTC0 the next day, and also inconsistent comparisons across political borders where the timezone changes. Figure 7b highlights this timestamp shifting by plotting the unshifted precipitation amount from AWAP averaged across Australia during cyclone Yasi as a dashed line, and the shifted AWAP and REGEN estimates as solid lines. Note that around 10% of observations in the US are midnight observations, i.e. observations over the 24h period from midnight





to midnight which are assigned to the day on which the observing period ends. Although these observations have not been manually adjusted in this version of REGEN, they will be taken care of in the next iteration.

The merging algorithm used is described below. Two stations were considered identical if:

1. The latitude and longitudes matched to three decimal places, and their elevation (to the nearest integer, if non-missing) and World Meteorological Organisation (WMO) station IDs either match or are missing. Alternatively the stations were also considered a match if the WMO IDs were non-missing and matched and the latitude and longitude matched to one decimal place.

2. If the coordinates were within 1 degree of each other and WMO IDs either matched or were missing and the correlation between the timeseries that overlap was greater than 0.99 and the overlapping timeseries themselves had at least 365 daily data records with a minimum of 10 days with precipitation greater than 1mm.

Note that the above algorithm can result in false matches as nearby stations can be highly correlated, however this will mainly be an issue in higly dense networks such as US. For the future version, a more quantitative measure of similarity between station time series will be used. On occasions where precipitation amount from a station was different between multiple sources, we prioritised data from higher quality sources and accepted values from these sources. The data qualities and hence priorities in descending order (highest quality first) are GPCC data base, GPCC ASCII data files, Other data, GHCN-Daily data.

## 2.4 Interpolation Method

Station data were interpolated using ordinary block Kriging, exactly as the method used by GPCC's Full Data Daily V1 (GPCC-FDD1; (Schamm et al., 2015)) product. Ordinary block Kriging is a stochastic interpolation method which means it accounts for the statistical structure of precipitation in terms of the spatial autocorrelation function. The autocorrelation function models the statistical relationship between the euclidean distances between the observations and their correlation. The interpolation method calculates a weighted average of the nearest station values based on their distance to the grid point and the autocorrelation function. This interpolation method was chosen by Schamm et al. (2014) after a comparison with various different methods. It produces area average precipitation estimates implicitly by estimating the interpolated field at various points inside the grid box and then calculating their weighted sum. This results in estimates directly comparable to other forms of data that produce area average estimates such as satellite products or climate models. More details of the interpolation method, including the autocorrelation function and its parameters, equations to calculate kriging estimates and their numerical implementation are described in Schamm et al. (2014) and Rubel (1996).

We interpolated ratios of the daily precipitation to the total monthly precipitation. If both the daily records and monthly totals were zero, the ratio was set to zero as well to ensure consistency with monthly datasets. The absolute values were retrieved post interpolation by superimposing the interpolated ratios on the GPCC Full Data Monthly V7 product (Schneider et al., 2015). This dataset was chosen because it is a well established dataset recommended for historical precipitation, global water cycle and trend analysis (Becker et al., 2013; Schneider et al., 2014, 2017). Furthermore, GPCC-FDD1 also calculates ratios using this dataset and it was readily available on the GPCC High Performance Computer (HPC) where the interpolation was





performed. This approach is commonly known as climatology aided interpolation (CAI) and has two advantages. Firstly CAI reduces the influence of elevation and other variables (Hofstra et al., 2008) which allows us to interpolate with only latitude and longitude as input variables. Secondly, because monthly gridded datasets are often based on much more reliable and stable station networks, especially in areas with problematic daily station coverage, the final absolute values may be more reliable

in these regions. The monthly totals for calculating daily ratios in the station timeseries were obtained by summing the daily station data as well. A month was considered complete if it had at least 70% of non-missing days. This, however, was a disadvantage of interpolating anomalies, as even if a daily record existed, it was not used for interpolation if the monthly total was missing because of the completeness criteria. Finally since we use GPCC Full Data Monthly V7 to retrieve daily absolute precipitation values, our analysis is also limited to the temporal extent of this monthly dataset which is currently up to the

year 2013. The interpolation parameters and auto-correlation function were also identical to the GPCC-FDD1 product and are described in (Schamm et al., 2014). The interpolation scheme uses the nearest 4 to 10 stations for interpolation (the numbers were chosen to have similar settings as the modified SPHEREMAP scheme utilised for the monthly analysis) and stations within 1 km are averaged to remove station duplicates as well as reduce the impact of such nearby stations on the estimate. For complete coverage, however, the search radius is increased until the minimum station requirement is met. This means that for

these stations in data sparse regions, the search radius can be much bigger than the decorrelation length scale of 347 km which is reflected in the Kriging error (see below). The decorrelation length scale is calculated from the autocorrelation function and is indicative of the extent of a station's influence.

Besides the interpolated fields, three other fields characterising the underlying data or uncertainty are provided with the dataset. These are

1. Kriging error which is not an absolute error but rather can be interpreted as percentage of variance (Rubel, 1996). It is a result of solving the Kriging equations and is dependent on the density of the observations and size of the grid (Schamm et al., 2014).

    2. Yamamoto standard deviation. This can be interpreted as an absolute error as it is the variance between the estimate and the observations used in interpolation, weighted by the Kriging weights (Yamamoto, 2000).

3. The field of number of stations inside each grid cell is also provided. Note that these are the actual number of observations inside a grid box. Note that this is not the number of stations used for interpolation of that grid cell estimate.

The 1950–2013 average Kriging error (KE) and coefficient of variation (CoV), and the data quality mask based on KE and CoV are shown for REGEN and REGEN40YR in figure 5. The CoV, defined as the ratio of the Yamamoto standard deviation and the precipitation estimate, is a normalised measure of the variance at each grid cell. The Kriging error is largest in regions

with a low station density such as Greenland, Africa and South America and is larger for REGEN40YR compared to REGEN as expected (figures 5a and 5b). Coefficient of variation, however, is comparable between REGEN and REGEN40YR. The largest CoV values are once again seen in Africa, South America, Greenland and Southeast Asia (figures 5c and 5d). The resulting data quality mask based on Kriging error and coefficient of variation for REGEN40YR has a smaller global land coverage with particularly sparse coverage in Africa, South America and Asia in both version of the dataset (figures 5e and 5f).



As mentioned earlier, we interpolated two different sets of underlying station data to create two related datasets. The first interpolates all available station data while the second interpolates only the long term data defined by stations with at least forty complete years of data, where a year was considered complete if all twelve months were non-missing, i.e each month had at least 70% non-missing days. The All station dataset (REGEN) is useful for those users who do not have access to a

regional precipitation product based on a high station density and would like an approximate estimate of precipitation as well as for users interested in the best estimate (based on as many stations as possible) of precipitation amounts at each time step, accepting that this may result in a decrease in temporal homogeneity. It is also useful for users seeking more complete fields of precipitation over global land areas and less interested in the uncertainties introduced due to station network variability. REGEN40YR is useful for users conducting a climate scale analysis of precipitation such as looking at trends in various

precipitation indices over several decades, since the use of long term stations minimises artificial variability of grid cell values due to network variations. We highly encourage users to use a dataset (REGEN or REGEN40YR) that is suitable to their needs in conjunction with a quality mask (described below).

We provide a quality mask for both datasets where the masked grids are of lower quality. The masks were prepared based on the Kriging error and coefficient of variation. Figures 5e and 5f shows the data quality masks for the two REGEN datasets.

A grid cell was left unmasked if it either contained at least 60% days in every decade from 1950 to 2013 (7 in total) with at least one station or both the grid cell coefficient of variation and Kriging error were under the 95$^{th}$ percentile threshold of the 1950–2013 average coefficient of variation and 1950–2013 average Kriging error respectively. For ease of use we provide a single mask for the entire data period, however, we recognise that the coefficient of variation, Kriging error and number of stations per grid vary over time, meaning a different mask could be calculated for each day. Such a mask would keep all grid

cells with at least one station in addition to all grid cells with the coefficient of variation and Kriging error within the 95$^{th}$ percentile of all the grids on the day. A possible recommended use case for the unmasked (high quality) grids of REGEN would be the evaluation of or comparison with another dataset (such as a satellite product) or climate model output.

## 3   Results and Evaluation

In this section we evaluate REGEN and REGEN40YR with existing monthly and daily precipitation datasets by showing

comparisons of maps and timeseries.

### 3.1   Comparison with global gridded datasets of monthly precipitation

Traditonally, global trends in historical precipitation are analysed with monthly datasets since no other suitable long-term daily datasets existed (e.g. Hartmann et al. (2013)). Here we reproduce the trend comparison from Hartmann et al. (2013) while including REGEN. Annual precipitation anomalies are compared in figure 6 between REGEN, REGEN40YR, GPCC Full

Data Monthly Version 7 (GPCC; (Schneider et al., 2015)), CRU TS v4.01 (CRU; (Mitchell and Jones, 2005)) and GHCN Monthly Version 2 dataset (GHCN; (Peterson and Vose, 1997)). Anomalies were calculated by subtracting the average of total annual precipitation from 1950–2010 from the total annual precipitation for each dataset respectively. The variability in annual



precipitation totals between REGEN and the other datasets is very similar, especially when compared to GPCC-FDD1 and CRU. GHCN has higher variability in many years compared to the other datasets including REGEN and REGEN 40YR.

## 3.2 Comparison with regional gridded datasets of daily precipitation

Regional gridded datasets of daily precipitation are often created by local research organisations and as such are often based on a much denser station network than global datasets (compare for e.g. Herrera et al. (2012) and Haylock et al. (2008)) and an interpolation method optimised for the local regions. The result is a dataset with long temporal records ideal for analysing individual events and precipitation extremes. REGEN's skill in capturing individual significant precipitation events is highlighted by a comparison of timeseries of daily totals from various events between REGEN, REGEN40YR and other commonly used regional datasets available for Europe, USA, Australia and Asia (figure 7). There is good agreement between the daily timeseries from REGEN, REGEN40YR and both 0.25 degree and 1 degree versions of E-Obs Version 16 (Haylock et al., 2008) for the events of the "Great flood of 1968" in Southeast England and France (Jackson, 1977) (figure 7a). Precipitation shown is spatially averaged over Ireland, Southern England, Northern France, Belgium and Netherlands with the events occurring in mid-September. In Australia, the precipitation events around the landfall of Cyclone Yasi in 2011 are compared between REGEN, REGEN40YR and the Australian Water Availability Project (AWAP) (Jones et al., 2009) dataset which is the most commonly used dataset of daily precipitation. Since the *in situ* data for Australia was shifted a day back during the production of REGEN, the AWAP daily averages were also shifted a day backward for this comparison and the agreement is high between the three datasets. Similarly, daily precipitation timeseries averaged over the Philippines during the Tropical storm Thelma in 1991 are shown in figure 7c. In this case we compare REGEN and REGEN40YR against APHRODITE (Yatagai et al., 2012) which is the longest running freely available dataset of daily precipitation in Asia at the moment. REGEN and especially REGEN40YR contain a lot fewer stations compared to APHRODITE in this region (figure 7c) which results in much larger differences in estimates between the datasets (figures 7a and 7b). REGEN captures the daily variability in APHRODITE well on most days however the long term version (REGEN40YR) with a lot fewer stations (due to the strict completeness criteria) exhibits larger differences, substantially underestimating the daily rates around October 27[th] and November 5[th] and overestimating on November 1[st] and November 9[th]. Finally based on a comparison of daily rainfall rates during tropical storm Amelia that made landfall in southern United States, there is also good agreement between REGEN, REGEN40YR and CPC CONUS (Chen and Xie, 2008; Xie et al., 2007; Chen et al., 2008) (figure 7d).

## 3.3 Case study over Sub-Saharan Africa

Based on the maps of Kriging error (figures 5a and 5b) the most data sparse regions of REGEN are Africa, South America, Greenland and northern Russia. Despite the sparsity of data, REGEN can still be useful to get estimates of daily rainfall in some parts of these regions. We use the country of Benin in sub-Saharan Africa as an example. Benin has a tropical climate receiving the majority of rainfall around the summer months of June-August (JJA). In the summer of 2008 Benin experienced catastrophic flooding events displacing around 150,000 people (WHO, 2010). The flooding started with heavy rainfall in the last week of July (IRIN, 2008). The timeseries of daily rainfall from 1950 to 2013 highlights 2008 as a year with the third



highest rainfall on record based on REGEN (figure 8a) with the highest being in 1957. On comparison of the daily rainfall timeseries between 1957 and 2008 (figure 8b), the anomalous rainfall in late June and late July is apparent, even compared to 1957. This highlights REGEN's effectiveness in capturing the daily rainfall even in some parts of Sub-Saharan Africa. Note that the region of Benin is of higher quality compared to surrounding regions as it is not masked in the data quality mask (figure

5e).

## 3.4    Comparison with existing global datasets of daily precipitation

Finally, in this section the only other existing global gridded gauge-based datasets of daily precipitation are compared. The temporally averaged annual total, annual maximum precipitation, trends in annual total and trends in annual maxima are compared against NOAA Climate Prediction Center's (CPC) Unified Gauge-Based Analysis of Daily Precipitation (CPC-Global)

(Chen and Xie, 2008; Xie et al., 2007; Chen et al., 2008) and GPCC Full Data Daily V1 (GPCC-FDD1). For comparability CPC-Global whose native resolution is 0.5 degrees was regridded to 1 degree to match the GPCC-FDD1 and REGEN. The temporal coverage of CPC-Global and GPCC-FDD1 is 1979–2017 and 1988–2013 respectively. The temporal averaging and comparison was therefore done over 1988–2013 which is the longest common period between the three datasets. As expected REGEN is more similar to GPCC-FDD1 and REGEN40YR compared to CPC-Global for both the means and trends of both

indices. This is because REGEN and GPCC-FDD1 use the same interpolation method and for the most part even the same underlying data. The largest differences between the three datasets arise in data sparse regions in the high latitudes, Africa, South East Asia, and the high altitude regions in western South America. The spatial variability of the differences in annual total and annual maxima trends is higher compared to the spatial variability of differences in averages of the annual totals and annual maxima. Due to the lack of long term stations in Saharan Africa differences in all four indices between REGEN and

the long term station based REGEN40YR are larger compared to differences between REGEN and GPCC-FDD1 in northern Africa. Herold et al. (2016) showed CPC-Global produces lower annual totals compared to an ensemble of observational datasets including GPCC-FDD1, satellite products and reanalyses. This is consistent with our results since the difference in annual totals between REGEN and CPC-Global are positive in majority of global land areas with the exception of northern North America and northern Africa.

Temporal and spatial correlation between REGEN and GPCC-FDD1 (figures 11a and 11b) are also higher compared to temporal and spatial correlation between REGEN and CPC-Global (figures 11c and 11d). In fact the spatial and temporal correlation between REGEN and GPCC-FDD1 is even higher than the correlation between REGEN and REGEN40YR (figures 11e and 11f) because REGEN's station network is more similar to GPCC-FDD1 than REGEN40YR. The areas with poor temporal correlation between REGEN and REGEN40YR correspond to areas with low station density such as the high latitudes,

Africa and South America. Compared to the field correlation between REGEN and GPCC-FDD1, the correlation between REGEN and REGEN40YR is also more variable. This may be because the lower station density results in an increase in daily variability in interpolated fields. The drop in field correlation between REGEN and GPCC-FDD1 around 2010 corresponds to the higher percentage of GHCN stations in the last years (figure 1b). There is also a decline in field correlation over time between REGEN and REGEN40YR which may be related to the decline in the number of long-term stations over time. The



temporal correlation between REGEN and CPC-Global is highest in USA, Australia, East Asia and a small part of Europe. These regions all correspond to regions with good station density throughout the time period.

# 4    Summary, limitations and best practice recommendations for users

We present a new gauge-based dataset of gridded daily precipitation with a grid resolution of $1° \times 1°$, global land coverage, and temporal coverage from 1950 to 2013 called REGEN. REGEN was produced by interpolating quality controlled *in situ* daily rainfall timeseries data using ordinary block Kriging. The interpolation method for REGEN is identical to GPCC-FDD1 (another gridded dataset of daily precipitation from 1988–2013). REGEN also uses all the *in situ* daily data used by GPCC-FDD1 but expands on this raw data by combining it with GHCN-Daily and raw data from other sources. This resulted in an extended *in situ* daily precipitation network with coverage back to 1950. The raw data were subjected to comprehensive automated control procedures identical to the one used by the GHCN-Daily dataset and all suspicious data were removed, interpolating only the high quality data. We used climatologically aided interpolation (CAI) which involved interpolating ratios of daily totals and monthly totals and retrieving absolute values by superimposing gridded monthly precipitation fields on the interpolated anomalies. This approach results in more reliable estimates in regions with sparse daily *in situ* data network and a comparatively denser monthly *in situ* data network. CAI also reduces the influence of variables such as elevation, distance to the coast etc. which allows us to interpolate using only the latitude and longitude as input variables. The gridded monthly fields used to retrieve the absolute daily precipitation rates came from GPCC Full Data Monthly V7 dataset.

REGEN is currently the longest running dataset of daily precipitation based on gauge-only records with global land coverage making it ideal for any global analysis at climatological scales. We therefore hope it will contribute to the advancement of hydrological science and practice by enabling a number of studies aiming to understand changes and variability in several aspects of daily precipitation distributions, including precipitation extremes, and measures of hydrological intensity. So far the only datasets that allowed global climatological scale analyses of precipitation were monthly datasets or gridded ETCCDI indices, however, the monthly datasets tend to average out the extremes, in turn losing their usefulness when it comes to high impact phenomena related to intense rainfall at shorter timescales. REGEN due to its daily temporal resolution fills this data gap. REGEN like GPCC-FDD1 also provides various uncertainties related to the daily gridded fields which include the Yamamoto standard deviation which is indicative of the proximity of the estimated fields to the raw stations values, the Kriging error which is indicative of the density of stations inside the grid cell and finally also the exact number of stations inside each grid cell. Based on these measures a quality mask for REGEN that combines all three uncertainty information indicating the high quality grid cells (with low uncertainties) is also presented. Users of REGEN are encouraged to use the quality mask in all cases except when spatial completeness is of utmost importance. Alongside REGEN (that interpolates all station data) another related dataset that minimises artefacts due to station network variability by interpolating only the long-term stations (i.e. stations with at least 40 years of complete data) is also produced. Both datasets include bespoke data quality masks. As a result, although the station density is lower in the long-term version, users can use its quality mask to restrict their analysis to higher quality areas. For analyses sensitive to the station network variability the long term station version with the high quality



mask would be the most suitable. Note, however, due to the lower station density, the long term station version may be less suitable for investigating individual events or short timeseries. The All station version on the other hand would be more suitable for analysis where a complete global coverage is important but temporal homogeneity is of lower priority. In any analysis it is recommended to use the data quality mask, however, in regions where no other daily datasets are available (such as parts of
Africa), REGEN may provide a suitable rough estimate of precipitation even in lower quality grids.

REGEN has been compared with global monthly and daily, and regional daily gridded datasets of precipitation. The annual precipitation anomalies have been shown to resemble those from the other monthly datasets and the spatial fields of annual totals and maxima as well as their trends more closely resemble GPCC-FDD1 than CPC. Even the daily timeseries of individual events of significant precipitation resemble the respective regional datasets closely in Europe, Australia and USA. The larger
inconsistencies between the long term REGEN data and APHRODITE in Asia are indicative of the lower station densities in REGEN in this region. Also note that there is almost no raw *in situ* daily data in mainland China in 1950. As such any analysis focusing on China using this dataset should not go further back than 1951. Finally, note that despite our best efforts to homogenise station data before interpolating, because the raw data are sourced ultimately from various countries with different measurement practices (such as time of measurement, use of units, quality control and homogenisation steps etc.),
inhomogeneities across political borders are possible (Trewin, 2010).

The biggest strength of REGEN is the long temporal coverage of quasi-global daily precipitation observations. Regional datasets are often developed by national meteorological organisations and often have access to significantly more data than shared with Global archives such as GHCN-Daily and GPCC. For example the Spanish dataset (Herrera et al., 2012) uses roughly the same number of stations just in Spain as E-OBS does for the entirety of Europe. Furthermore, Herrera et al. (2012)
only used the high quality stations which accounted for roughly 30% of total stations available from the Spanish Meteteoro- logical Agency (AEMET). Often the respective meteorological organisation also have the resources to more thoroughly and in some cases even manually quality control the raw data. As a result, regional datasets (where available) may provide more accurate precipitation estimates than REGEN.

At the moment REGEN is not an operational product, meaning the analysis for REGEN was done as a single instance and
there are currently no plans on updating it regularly, such as on an annual or biennial basis.

Figure 12 reflects REGEN's strengths by showing annual totals and maxima and trends over the high quality regions over the entire 63 year record of REGEN. Both the total annual precipitation and annual maxima based on REGEN are reasonable with higher totals and maxima in the known wet regions such as the tropics and lower totals and maxima in the known dry regions such as Saharan Africa (figures 12a and 12b). Trends in total precipitation based on REGEN (figure 12c) are also
comparable to the trends in total precipitation shown in the IPCC's 5[th] Assessment report (figure 2.29, Hartmann et al. (2013)). The total annual precipitation, annual maxima and respective trends in the two indices based on the long term REGEN data (REGEN40YR) (figures 12e, 12f, 12g and 12h) are also very similar to REGEN which suggests that the effects of station variations appear negligible at this scale (for trends and averages over 1950–2013) for the high quality grids. The trend maps shown in figure 12 have been masked based on the quality masks as shown in figures 5e and 5f.





REGEN has proved itself by providing precipitation estimates comparable to those from the currently most reliable datasets such as GPCC-FDD1. With a temporal coverage $152\%$ longer than that of GPCC-FDD1's and a similar global land coverage, REGEN is highly suitable for analysing climate change. We recognise that observations are not the "truth" but rather just our best estimates of it. To this note, REGEN and its variant REGEN40YR which minimises station network variability, are

accompanied by various uncertainty estimates as well as a quality mask, allowing users a firm handle of the observational uncertainties in their analysis.

*Data availability.* REGEN and REGEN40YR data has now been published with unique Digital Object Identifiers (DOIs) https://dx.doi.org/ 10.25914/5b9fa55a8298c and https://dx.doi.org/10.25914/5b9fa67fce5d6 respectively. Both datasets can be acquired in netcdf format along with netcdfs of the quality masks via the Research Data Australia (RDA) web pages https://researchdata.ands.org.au/rainfall-estimates-gridded-station-v10

1340689 and https://researchdata.ands.org.au/rainfall-estimates-gridded-station-v10/1340690 respectively. The RDA records contain further information about the datasets such as the dataset abstract, citation information, related organisations, grants, researchers and dataset managers (SC).

## Appendix A: List of countries for which the timestamps have been shifted a day back

The countries for which the data are shifted a day back (e.g. data from $2^{nd}$ Jan are saved as $1^{st}$ Jan) are Australia, Bangladesh,

Brazil, Benin, Bulgaria, Costa Rica, Denmark/Greenland, Georgia, Indonesia, Japan, Kenya, Netherlands, Norway, Croatia, Slovenia, Suriname, Turkey, Ukraine, Angola, Antarctica, Argentina, Azerbaijan, Bahamas, Barbados, Bolivia, Botswana, Burkina Faso, Cameroon, Chad, Chile, Ethiopia, Gabon, Georgia, Guam, Hungary, Indonesia, Islands in the Indian Ocean (IOT), Ivory Coast, Libya, Madagascar, Malawi, Mali, Marshall Islands, Mozambique, Mauritania, Niger, Peru, French Polynesia, Sudan, Senegal, Solomon Islands, Tunisia, Tanzania, Uruguay, Vietnam, Vanuatu, Zambia and Zimbabwe.

*Competing interests.* The authors declare that there were no competing interests regarding the publication of this article

*Acknowledgements.* This study was funded by Australian Research Council (ARC) grant DP160103439. LVA is also funded by ARC grant CE110001028. We are also grateful to the National Centers for Environmental Information (NCEI) for providing QC scripts, raw GHCN-Daily data and hosting SC on a research visit, and we are grateful to the Global Precipitation Climatology Centre (GPCC) for providing interpolation scripts, computational resources and for hosting SC on a research visit to allow for an on-site access to its data archive.





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





**Figure 3.** Grids showing percentage of days with at least 1 station in each decade for (figure 3a) REGEN, (figure 3b) REGEN40YR and (figure 3c) GPCC-FDD1. Gray areas indicate grids where no stations are present.




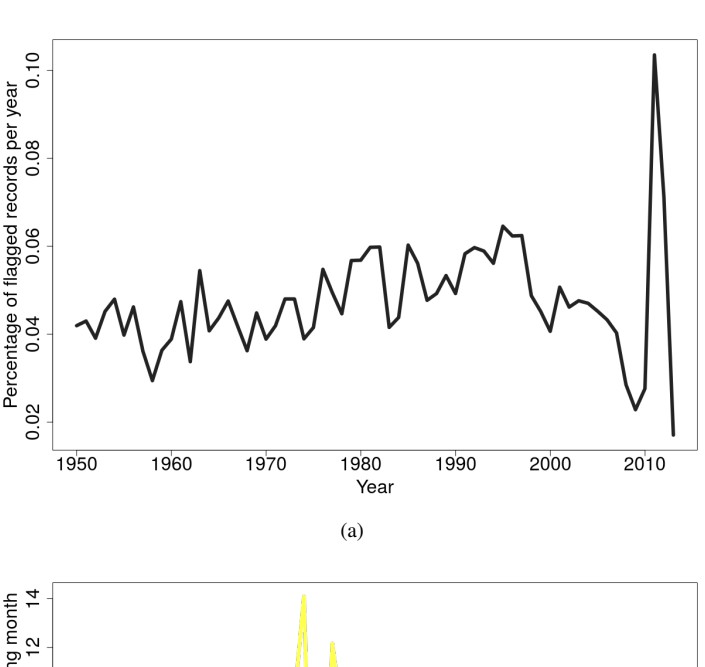

(a)

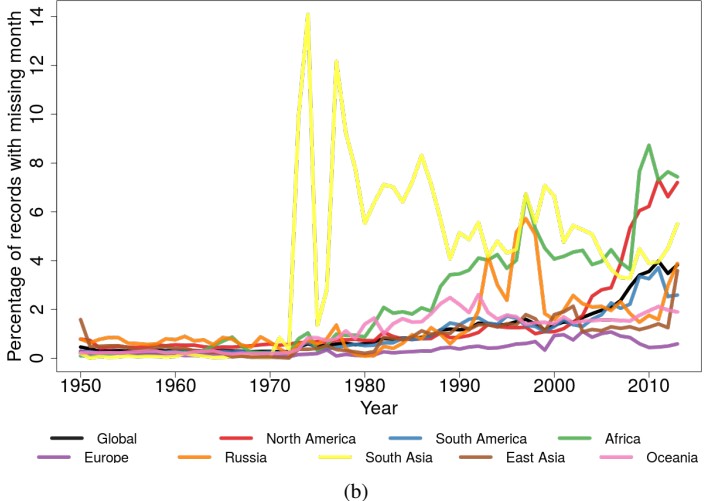

(b)

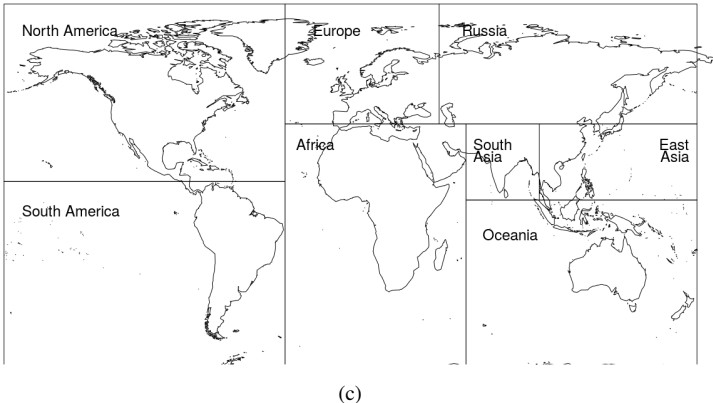

(c)

**Figure 4.** Percentage of records that (figure 4a) failed one or more quality control tests and were flagged and (figure 4b) were not used as input for interpolation due to missing monthly totals and hence missing anomaly values. Figure 4c shows a map of regions as used for figures 4b and 1a.







**Figure 5.** Kriging error (KE)(figures 5a and 5b), Coefficient of variation (CoV) defined by the ratio of the Yamamoto standard deviation (Yamamoto, 2000) averaged over 1950-2013 and the daily precipitation averaged over 1950-2013, and Masks based on the KE and CoV (figures 5e and 5f) based on REGEN (left Column) and REGEN40YR (right column) data.





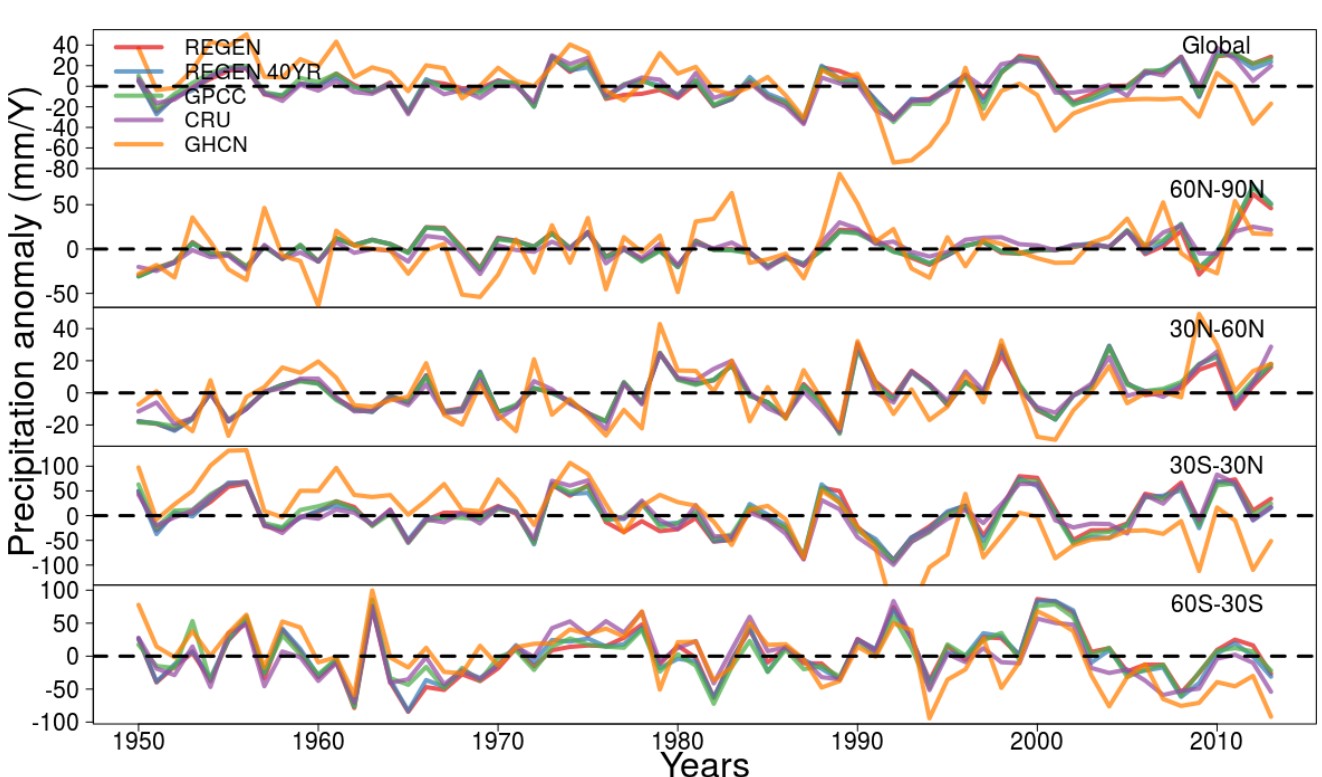

**Figure 6.** Comparison of annual precipitation anomaly timeseries with monthly datasets. Anomalies were calculated relative to the average of daily precipitation totals over the entire time period (1950-2013) for each dataset.







**Figure 7.** Daily timeseries avearged over spatial regions of significant rainfall events. (figure 7a) Timeseries of daily rainfall during the great flood of 1968 over Southeast England. (figure 7b) Timeseries of daily rainfall during Cyclone Yasi in northeast Australia in 2011. (figure 7c) Timeseries of daily rainfall during typhoon Thelma in Philippines in 1991. (figure 7d) Timeseries of daily rainfall during tropical storm Amelia in US in 1978.





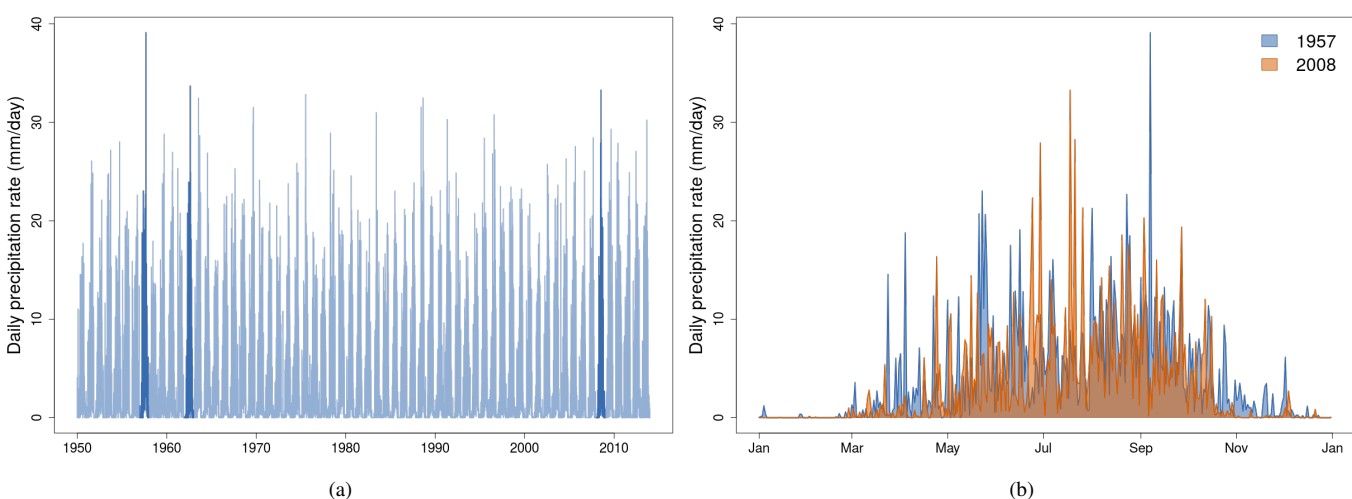

**Figure 8.** Timeseries of daily precipitation from REGEN averaged over Benin in Western Africa. Figure 8a shows the entire timeseries from 1950 to 2013 with the years containing the days with the highest three daily rainfall rates (1957, 1963 and 2008) shown in a darker shade. Figure 8b shows a comparison of the timeseries of daily rainfall between 1957 (year containing the day with the record highest rainfall based on REGEN) and 2008 (year during which the 2008 Benin floods occurred). Benin was chosen because of its good coverage of stations.





**Figure 9.** Percentage difference in averaged total annual precipitation (first column; figures 9c, 9e and 9h), averaged maximum annual precipitation (second column; figures 9d, 9f and 9g) between REGEN and GPCC (second row), REGEN and CPC (third row) and REGEN and REGEN40YR (fourth row) data. The first row shows the absolute values of total annual precipitation (figure 9a) and RX1DAY (figure 9b) averaged over 1988 - 2013 (the longest common time period between the three datasets).





**Figure 10.** Percentage difference in total annual precipitation trends (first column; figures 10c, 10e and 10g), and annual maximum precipitation trends (second column; figure 10d, 10f and 10h) between REGEN and GPCC (second row), REGEN and CPC (third row) and REGEN and REGEN40YR (fourth row) data. The first column shows the absolute values of total annual precipitation trends (figure 10a) and annual maximum precipitation trends (figure 10b) averaged over 1988 - 2013 (the longest common time period between the three datasets).





**Figure 11.** Spatial (field) correlation at each daily time-step (first column; figures 11a, 11c and 11e) and temporal correlation between timeseries at each grid cell (second column; figures 11a, 11c and 11e) between REGEN and GPCC (first row), REGEN and CPC (second row) and REGEN and REGEN40YR Long term (third row) data.





**Figure 12.** Total annual precipitation (figures 12a and 12e), maximum annual precipitation (figures 12b and 12f) and respective trends (PRCPtot; figures 12c and 12g and RX1DAY; figures 12d and 12h) averaged over 1950 to 2013 based on REGEN data (figures 12a, 12b, 12c and 12d) and REGEN40YR data that only interpolates stations with at least forty complete years of data (figures 12e, 12f, 12g and 12h).