# Peer review of "Rainfall Estimates on a Gridded Network (REGEN) - A global land-based gridded dataset of daily precipitation from 1950–2013"

_Hydrology and Earth System Sciences, 2018_

## Referee Comment (RC1) · Anonymous Referee #1 · 13 Mar 2019

review of "Rainfall Estimates on a Gridded Network (REGEN) - A global land-based gridded dataset of daily precipitation from 1950-2013" by Steefan Contractor et al.

The paper presents and documents a global daily precipitation dataset at a 1 degree resolution. The dataset is compiled from several data sources. In the paper, discussions on the method to grid daily precipitation and the density of the network, including the consequences this has for the precipitation estimates on the grid, are included. In addition to REGEN, a dataset based only on the long-running stations is presented which will be temporally more homogeneous than the dataset based on all station data.

The study is a very welcome contribution to the field where for daily precipitation now

many national and several regional datasets exist, but (up to now) not a global dataset based on in-situ measurements. The study is well written and clear and as far as I can judge, no problems in the analysis are there. However, the study could do with a more expansive and in-depth comparison against existing (regional) observational datasets - the current comparisons are too ad-hoc and uninformative.

My advise to the editor is to accept the paper with minor adjustments.

**More serious concerns**

1. Metadata is often a problem (lacking, erroneous etc.). The day shift discussed in sect. 2.3 - a very necessary thing to do - is in the face of poor metadata an action which might be problematic. The give an example, according to the Appendix, the data from the Netherlands are shifted one day backward in time. This is appropriate for the manual rain gauges but not for the 30+ automatic weather stations which measure precip between 0-0 UTC. It could be that only the rain gauges are in the GPCC dataset - but the reader can't tell.

   Checking with other NMHSs in Europe, I could confirm the necessity to shift the date, except for Hungary (there is a question to the Hungarian NMHS out now).

   There is another confusing part of this date-adjustment. Are you aiming to get 24-hour values coinciding with a day defined by local time or by UTC? The reason for asking is that for Indonesia, my understanding is that measurements are from 7 - 7 local time, which is (nearly) O - O UTC.

   The correlations with CPC (figure 11d) are low in some areas - could it be that an erroneous timeshift or a missed time shift could be related to the low correlation? I guess you have tried to shift the whole dataset back and forth and looked for areas on the globe with increases in correlation?

2. The comparison against regional dataset of daily precipitation (sect. 3.2) is too ad-hoc. In your article, you claim (rightly so) that national and regional datasets

are based on a more extensive dataset. I would like to add that especially national datasets have a far greater detail in the understanding of the metadata. This means that a meaningful comparison can be made between national/regional datasets and the REGEN dataset. This should go beyond simply picking one event of a few days, averaging precip over a region and plot a few timeseries.

Please add a more expansive comparison with regional datasets like Aphrodite etc. Other datasets might be interesting to use as well, like the SA-OBS for Southeast Asia (van den Besselaar et al. 2017. doi:10.1175/JCLI-D-16-0575.1). Comparison you could make easily are the standard deviation of the daily difference, perhaps stratified over different periods, but other comparison metrics are equally useful. Given the particular focus on precipitation extremes by the international community - and of some of the authors of the paper - a dedicated focus of representation of extremes (beyond one example) is required.

I was taken by surprise when reading about the the "Great flood of 1968" in Southeast England and France. The article referred to (Jackson, 1977) never mentions France. Below are a few pictures from the E-OBSv19.0e for 14-16 September 1968 and the area you use to compare REGEN with the regional dataset is somewhat large compared to the area affected by this event.

**Other issues the authors may want to look into**

1. page 2, line 16: here is is claimed that radar provides 'highly accurate' estimates of precipitation. It is my understanding that radar can underestimate extreme precipitation by as much as 40% (e.g. https://journals.ametsoc.org/doi/10.1175/2009BAMS2747.1)

2. the referencing to figures is a bit curious: The first 3 are referenced chronologically, but then on page 7, you refer to fig. 4b, the next reference (line 31) is to fig. 7bpage 9 10 have refs to fig 5 and page 11 has a reference to fig 7.

[Figure]

3. page 8, lines 4-7. Relocated stations often keep the WMO id and if the relocation is to a site in the vicinity, then your criterion labels the old and new station as the same. This may not be a problem for precipitation, but perhaps it is good to inform the reader about this.

4. page 11, line 10. There is no 1.0 degree version of the E-OBS. I guess you regridded the E-OBS data to the REGEN grid to arrive at the 1 degree resolution?

5. page 14, line 19-20. Here you make the point that there is an ordering in the number of stations used by national, regional and global datasets. The point is very valid, but the example provided is misleading. Herrera used 2756 stations, the E-OBS uses 210 station in Spain (incl. Catalonia) and for the whole of Europe, 15962 series are used. Hardly 'roughly the same number' as claimed.

**Very minor issues**

1. page 3, line 4, It is the Climatic Research Unit (not Climate)

2. page 7, line 4, typo in procedures

3. page 9, line 20, perhaps an odd formulation?

4. references, many citations have the http address twice in the citation, e.g. Jackson (1977).

5. Appendix A. you apparently made an effort to make an alphabetical list - but didn't quite succeed. There are duplicates in the list too - like Indonesia.

6. caption fig. 5: fig. c  d show what?

Daily precipitation amount (rr) (rr)

**Fig. 1.** rainfall for 1968/09/14 based on E-OBSv19.0e

Daily precipitation amount (rr) (rr)

**Fig. 2.** rainfall for 1968/09/14 based on E-OBSv19.0e

Daily precipitation amount (rr) (rr)

**Fig. 3.** rainfall for 1968/09/14 based on E-OBSv19.0e

---

## Referee Comment (RC2) · Anonymous Referee #2 · 30 Apr 2019

This paper presents a new global gridded rainfall dataset that combines existing gauge datasets, quality controls the data and produces a useful new gridded daily rainfall product. Rainfall data at a daily timestep are difficult to access and in parts of the world, not even collected. The work presented here represents a substantial effort and contribution to both the literature on rainfall data and in providing a new resource for hydrologists amongst others. Well done!

The paper is well written and the methods and analysis are sound. I think the paper only requires some minor revisions to be acceptable for publishing.

P2 L9: ALL measurements of PPT have errors including gauges. Please clarify this

and add in a short discussion of the errors in gauge data. (see McMillan et al 2012 as a starting point)

P2 L10: Who thinks that reanalysis products represent the 'true state of the system'? Either cite or say that reanalysis data are sometimes misused as observations

P2 L13 NASA MERRA2 assimilates precipitation (https://gmao.gsfc.nasa.gov/pubs/docs/McCarty885.pdf Table 1)

P2 L16 There are significant errors associated with rainfall measurements from RADAR. I would not call then 'Highly accurate' (e.g. Krajewski et al (2010) gives a good summary of radar-rainfall uncertainties, focusing on improvements since the key paper by Wilson and Brandes in 1979. A more recent paper which focuses more on the applications in urban hydrology is Thorndahl et al (2017) lists a lot of uncertainties. Attempts to model the uncertainties are given in Villarani et al (2014), Rico-Ramirez et al (2015), Bong-Chul Seo and Krajewski (2015) and Cecinati et al (2017).)

P2 L19 Replace 'global if not quasi-global' with 'global/quasi-global'

P2 P3 I think that GPM should be mentioned as this is the most cutting edge satellite measurement of rainfall, Also mention blended datasets: MSWEP (https://journals.ametsoc.org/doi/full/10.1175/BAMS-D-17-0138.1), Blended gauge/satellite (https://agupubs.onlinelibrary.wiley.com/doi/pdf/10.1002/2013JD020686) etc.

P4 L15 Given that you are taking the GPCC to be the 'best' dataset, I would list that first.

P4 L15-18 It would be useful to put in brackets the number of gauges in each data source.

P4 L18 Include a table of 'other' data sources that can be referred to here. Currently left wondering for a long time what the other data sources are. Is it just Argentina and Russia? If so, just state that.

P4 L24 Re-phrase to be a call to action for met services to share more data.

P5 L3 Any idea why there was a decline in 2010?

P6 F2 Awesome! I can't really see any of the 'other' points though

P6 L7 Clarify if you used existing QC code or if you rewrote it based upon Durre et al.

P7 L1 Describe any validataion of the QC code that was undertaken to assess correctly/incorrectly flagged values.

P7 L11 Why did you choose 70% for a threshold?

P7 L13 Why is there a drop in Indian station data in the 70s?

P8 L12 'Higly' typo

P8 L8 1 degree seems like a huge area to consider gauges to be the same over. Why was such a large area used? How many gauges were merged in this way?

P8 L15 Please could you elaborate on the merging process. how were records combined? did you use the whole record from the highest quality source? or did you insert, for example, a few days from a GHCND record into periods of missing data from a GPCC record? If so, did you replace all missing days or did you only replace when there were a whole month of values? Do you have any idea of how many values were merged this way and whether it impacted the homogeneity of records?

P8 L29 Was it ever the case that the daily gauges showed no rainfall for the month but the monthly gauge did? If so, what did you do? Also, what do you do with one or more missing days of data? By disaggegating a monthly rainfall value with an incomplete daily record, you will most likely be increasing the average daily rainfall and reported extremes.

P8 L30 Please clarify how the monthly data is used. Is it the case that you are effectively temporally disaggregating the monthly rainfall to a daily timestep, ultimately

preserving the monthly values? Or will the monthly totals end up being slightly different?

P9 L25 I'm confused by point 3- how are the number of observations different to the number of stations? Is it that you may have 5 stations in the grid box but one of them has no data for that day and so the number of observations is 4? Please clarify.

P9 L27-35 Please add in some descriptive numbers of how 'wrong' the interpolated rainfall can be.

P10 L9 Replace 'trends' with 'changes' (we should discourage the use of trends in hydrology: https://doi.org/10.1016/j.advwatres.2017.10.015)

P10 L11 replace "we highly encourage uers to" with "Users must"

P10 L15-17 This is very unclear. You use 'either' but do not provide an 'or'. Please clarify.

P11 L2 Is this because of the QC applied to GPCC and REGEN?

P11 L9 Can you provide any estimates of how many more stations are used in national datasets compared to those used in REGEN?

P11 L10 State what interpolation method E-Obs used.

P12 L19 Comma after Africa

P13 L17 Remove 'running'

P13 L28 Replace 'are encouraged to' with 'should'

P14 L25 It is such a shme that REGEN will not be updated.

P14 Include a discussion of the limitations of 1 degree dataset. Rainfall is highly spatially variable and a ~100km2 estimate is unlikely to contain the information necessary for many typical rainfall aplications.

**HESSD**

P14 L31 Why should we expect differences in the total annual precipitation between REGEN and REGEN-40 if they have both been adjusted by the monthly data? Are the monthly totals not necessarily preserved?

P15 L1 Replace 'REGEN has proved itself by providing' with 'REGEN provides'

P15 L4 Remove 'To this note' ad include 'therefore' after are.

P15 L7 Include a statement about copyright/useage. Can anyone use this dataset freely? Industry? Or just for research?

FIGURES

Please include labels on all of your figures/scale bars. This makes them much easier to interpret than having to refer to the caption.

F4.c The lower line is missing.

F5 Caption: space needed between (KE) and (figures.... Missing '(figures 5c and 5d)' from the caption

F9 Please label the columns and rows on the diagram, it would make it much easier to interpret. I think 9h and 9g are mixed up in the caption. Also, Rx1day is not defined anywhere in the text or figures.

F10 Please label the columns and rows on the diagram, it would make it much easier to interpret.

Style points: - I found the italicisation of latin terms like 'in situ' distracting , especially in the abstract. - Why are you using 'in situ measurements' as opposed to 'gauge measurements' as your terminology?
* * *

---

## Author Comment (AC1) · 31 May 2019

**Response to anonymous referee #1**

The paper presents and documents a global daily precipitation dataset at a 1 degree resolution. The dataset is compiled from several data sources. In the paper, discussions on the method to grid daily precipitation and the density of the network, including the consequences this has for the precipitation estimates on the grid, are included.  In addition to REGEN, a dataset based only on the long-running stations is presented which will be temporally more homogeneous than the dataset based on all station data.

The study is a very welcome contribution to the field where for daily precipitation now many national and several regional datasets exist, but (up to now) not a global dataset based on in-situ measurements. The study is well written and clear and as far as I can judge, no problems in the analysis are there. However, the study could do with a more expansive and in-depth comparison against existing (regional) observational datasets - the current comparisons are too ad-hoc and uninformative.

My advise to the editor is to accept the paper with minor adjustments.

We thank the reviewer for their thoughtful and thoroughly researched comments and agree with the major criticism about the lack of a more in-depth comparision with regional datasets.

We have responded (in red) to each reviewer comment below. Page and line numbers refer to the original unmodified text.

**More Serious Concerns**

1.      Metadata is often a problem (lacking, erroneous etc.). The day shift discussed in sect. 2.3 - a very necessary thing to do - is in the face of poor metadata an action which might be problematic. The give an example, according to the Appendix, the data from the Netherlands are shifted one day backward in time. This is appropriate for the manual rain gauges but not for the 30+ automatic weather stations which measure precip between 0-0 UTC. It could be that only the rain gauges are in the GPCC dataset - but the reader can't tell.

Checking with other NMHSs in Europe, I could confirm the necessity to shift the date, except for Hungary (there is a question to the Hungarian NMHS out now).

There is another confusing part of this date-adjustment. Are you aiming to get 24-hour values coinciding with a day defined by local time or by UTC? The reason for asking is that for Indonesia, my understanding is that measurements are from 7 - 7 local time, which is (nearly) O - O UTC.

The correlations with CPC (figure 11d) are low in some areas - could it be that an erroneous timeshift or a missed time shift could be related to the low correlation? I guess you have tried to shift the whole dataset back and forth and looked for areas on the globe with increases in correlation?

We agree with the reviewer about the problems associated with poor metadata and the day shift discussed in the text. We are aware of automated weather stations that measure precipitation over a different 24h window to the country they are in. In the text we have identified a similar issue with 10% of American stations which were also automatic weather stations (P7 L33). We will expand the note on this issue as follows:

"Note that some countries maintain a mix of manually monitored and automated weather stations which may represent precipitation over differing 24h windows that may not be suitable for being

shifted identically. For example, around 10% of observations in the US and around 30 stations in the Netherlands are midnight observations, i.e. observations over the 24h period from midnight to midnight UTC which are assigned to the day on which the observing period ends. Although these observations have not been manually adjusted in this version of REGEN, they will be taken care of in the next iteration. Globally more countries may exist whose gauge observations may represent a mix of reporting times (due to the use of automatic weather stations for example), however, without proper metadata about these reporting times it is not possible for us to adjust their timestamp accordingly."

Regarding the hungarian data, it in fact has not been shifted by us. The inclusion of "Hungary" in the list of shifted stations in the appendix was an error. It will thus be removed from the list. We thank the reviewer for bringing this to our attention.

Dates are not adjusted to match 0-0 UTC time but rather to match the local 0700h-0700h local time to preserve the diurnal aspect of precipitation.

It is possible that correlations between REGEN and CPC are lower because of this shifting. Based on the correlation between REGEN V1.1 All Stations shifted +1 days and CPC (shown below) we see that the correlations are higher in and around the countries where data was shifted a day back (eg. Vietnam, Brazil, Uruguay, Peru, Suriname, Netherlands, Norway, Ukraine and Turkey). Lower correlations are observed in all other regions.

[Figure]

REGEN V1.1 lagged +1 days vs CPC_Glb temporal correlation 1988-2013

The above figure will be added to the supplementary materials (Figure S1) and the following note will be added to the text on P12 L26.

"Correlations between REGEN and CPC may be lower in parts where the underlying stations were shifted a day backward (see Appendix). Indeed, based on correlations between REGEN lagged +1 days and CPC (Fig. S1), the correlations are higher compared to figure 11d in and around the countries where data was shifted a day back (eg. Vietnam, Brazil, Uruguay, Peru, Suriname, Netherlands, Norway, Ukraine and Turkey). Correlations do not change compared to figure 11d in all regions where REGEN raw station data are not shifted."

2.      The comparison against regional dataset of daily precipitation (sect. 3.2) is too ad-hoc. In your article, you claim (rightly so) that national and regional datasets are based on a more extensive dataset. I would like to add that especially national datasets have a far greater detail in the understanding of the metadata. This means that a meaningful comparison can be made between national/regional datasets and the REGEN dataset.  This should go beyond simply picking one event of a few days, averaging precip over a region and plot a few timeseries.

Please add a more expansive comparison with regional datasets like Aphrodite etc. Other datasets might be interesting to use as well, like the SA-OBS for Southeast Asia (van den Besselaar et al. 2017.  doi:10.1175/JCLI-D-16-0575.1). Comparison you could make easily are the standard deviation of the daily difference, perhaps stratified over different periods, but other comparison metrics are equally useful. Given the particular focus on precipitation extremes by the international community - and of some of the authors of the paper - a dedicated focus of representation of extremes (beyond one example) is required.

I was taken by surprise when reading about the the "Great flood of 1968" in Southeast England and France. The article referred to (Jackson, 1977) never mentions France. Below are a few pictures from the E-OBSv19.0e for 14-16 September 1968 and the area you use to compare REGEN with the regional dataset is somewhat large compared to the area affected by this event.

We are currently in the process of adding a new figure that compares the mean and standard deviation of difference between REGEN and the regional datasets in the main text. These include those datasets already mentioned in the paper (CPC CONUS, E-Obs, Aphrodite, AWAP) as well as the SA-Obs dataset highlighted by the reviewer. The statistics can be aggregated over individual years and shown as a timeseries over the overlapping time period. In addition to the above figure we will also include maps of temporal correlations of local (grid-cell) timeseries between the regional datasets and REGEN to show a comparision of temporal variability between the dataset pairs. Finally, as suggested by anonymous referee #2, we will also include a table comparing number of stations between the regional datasets, REGEN and REGEN40YR now. This table is shown below.

| Regional Dataset Name | Regional Dataset Stations | All REGEN v1.1 stations | Long term (40yr) REGEN v1.1 stations |
|---|---|---|---|
| APHRODITE | Daily max of 8000+ | 8551 | 1539 |
| SA-Obs v1.0 | 7956 | 2527 | 64 |
| E-Obs | 17,468 | 28,338 | 11,261 |
| CPC CONUS | ~28,500 | 42,229 | 3940 |
| AWAP | Daily max of ~7500 | 12,993 | 1424 |

We will modify figure 7a so only UK is included in the spatial averaging and any mention of France can be removed. The updated figure is shown below.

[Figure]

**Other issues the authors may want to look into**

1.      page 2, line 16: here is is claimed that radar provides 'highly accurate' estimates of precipitation.  It is my understanding that radar    can underestimate extreme precipitation by as much as 40% (e.g. https://journals.ametsoc.org/doi/10.1175/2009BAMS2747.1)

The sentence will be replaced with the following:
"Radar estimates provide high spatial and temporal resolution estimates of rainfall over local regions, however these estimates can be inaccurate compared to rain gauges (Krajewski et al. 2010 Villarini and Krajewsky 2010 and McKee and Bins 2015), and very few national networks of radar observations exist.".

2.      the referencing to figures is a bit curious: The first 3 are referenced chronologically, but then on page 7, you refer to fig. 4b, the next reference (line 31) is to fig. 7b page 9 10 have refs to fig 5 and page 11 has a reference to fig 7.

Besides the reference to Figure 7b all other references are in order of figure appearance (see locations below). Figure 7b is referenced earlier as it helps to demonstrate the time-shifting of data. We leave it to the editor to clarify whether chronological figure referencing is strictly necessary, or whether referencing the Figure 7 once a little earlier is acceptable to improve readability.
Fig 4a,b: P6 L12, P7 L5
Fig 5a-f: P9 L26-27, P10 L10
Fig 6: P10 L25

3.      page 8, lines 4-7. Relocated stations often keep the WMO id and if the relocation is to a site in the vicinity, then your criterion labels the old and new station as the same. This may not be a problem for precipitation, but perhaps it is good to inform the reader about this.

The following note will be added to the text on P8 L8:
"Also note that WMO station IDs do not change after a station is relocated to a site in the vicinity which can result in two stations in different locations merged together according to our criteria."

4.      page 11, line 10. There is no 1.0 degree version of the E-OBS. I guess you regridded the E-OBS data to the REGEN grid to arrive at the 1 degree resolution?

The 1 deg version of E-Obs was created by regridding the 0.25 deg product using second order conservative remapping from CDO (cdo remapcon2). P10 L7 will be updated as follows to reflect this.
"There is good agreement between the daily timeseries from REGEN, REGEN40YR and both 0.25 degree and 1 degree (*regridded from 0.25 degree version using CDO remapcon2)* versions of E-Obs Version 16".
The figure label will also be updated (see figure above).

5.      page 14, line 19-20. Here you make the point that there is an ordering in the number of stations used by national, regional and global datasets. The point is very valid, but the example provided is misleading. Herrera used 2756 stations, the E-OBS uses 210 station in Spain (incl. Catalonia) and for the whole of Europe, 15962 series are used. Hardly 'roughly the same number' as claimed.

The sentence is false. We meant to say that the Spanish Meteorological Agency (AEMET) maintains roughly the same number of stations as those used in the entirety of Europe by E-Obs. The text (P14 L13) will be modified to reflect this as follows:
"For example, the Spanish Meteorological Agency (AEMET) itself manages roughly over 9000 stations (Hererra et. al. 2012) which is almost the same number of stations as those used by E-Obs for the entirety of Europe (around 12,000 gauges at its maximum)."

**Very minor issues**

page 3, line 4, It is the Climatic Research Unit (not Climate)
"Climate" will be changed to "Climatic"

page 7, line 4, typo in procedures
"prodcedures" will be changed to "procedures"

page 9, line 20, perhaps an odd formulation?
The sentence will be modified as follows:
"Kriging error. This is not an absolute error but rather…"

references, many citations have the http address twice in the citation, e.g. Jack- son (1977).
The repeated links will be removed from the following citations:

Yamamoto 2000, Xie et al 2007, Tian, Y. and Peters-Lidard 2010, Smith et al 2010, Schneider et al 2014, Schamm et al 2015, Peterson et al 1997, Perry and Hollis 2005, Osborne and Hulme 1998, Jackson 1977, Isotta et al 2013, Hofstra et al 2008, Herrera et al 2012, Harris et al 2014, Groisman et al 2005, Funk et al 2015, Frei et al 1998, Donat et al 2013b, Chen et al 2008, Bytheway and Kummerow 2013, Ashouri et al 2014, Allen and Ingram 2002, Alexander et al 2006, and Adler et al 2003.

Appendix A. you apparently made an effort to make an alphabetical list - but didn't quite succeed. There are duplicates in the list too - like Indonesia.

The countries will be re-ordered in alphabetical order. Also repeat entries of Georgia and Indonesia will be removed and "Guam" will be removed from this list as it is a US territory.

caption fig. 5: fig. c d show what?

The figure caption will be modified as follows:

"Figure 5. Kriging error (KE) (figures 5a and 5b), Coefficient of variation (CoV) (figures 5c and 5d) defined by…"

---

## Author Comment (AC2) · 31 May 2019

**Response to anonymous referee #2**

This paper presents a new global gridded rainfall dataset that combines existing gauge datasets, quality controls the data and produces a useful new gridded daily rainfall product. Rainfall data at a daily timestep are difficult to access and in parts of the world, not even collected. The work presented here represents a substantial effort and contribution to both the literature on rainfall data and in providing a new resource for hydrologists amongst others. Well done!

The paper is well written and the methods and analysis are sound. I think the paper only requires some minor revisions to be acceptable for publishing.

We thank the reviewer for the encouraging review. The suggestions and points raised help clarify any misleading information and provide additional detailed documentation of pertinent information related to REGEN.

We have responded (in red) to each reviewer comment below. Page and line numbers refer to the original unmodified text.

P2 L9: ALL measurements of PPT have errors including gauges. Please clarify this and add in a short discussion of the errors in gauge data. (see McMillan et al 2012 as a starting point)

The sentence on P2 L9 will be replaced with the following paragraph.
"All observations have errors, for example, gauge-based precipitation measurements are subject to undercatch, wind related errors, evaporation loss, wetting loss, splash in/out errors and tipping errors (see McMillan et al 2012 for details). However, alternatives to gauge-based measurements such as satellite observations, model reanalysis products and radar-based observations have additional limitations."

P2 L10: Who thinks that reanalysis products represent the 'true state of the system'? Either cite or say that reanalysis data are sometimes misused as observations

The sentence will be modified as follows:
"Reanalyses are often misused as observations but in fact inherit…"

P2 L13 NASA MERRA2 assimilates precipitation
(https://gmao.gsfc.nasa.gov/pubs/docs/McCarty885.pdf Table 1)

The sentence will be modified as follows:
"Furthermore, none of the reanalysis products assimilate gauge-based precipitation observations (MERRA2 however incorporates satellite infrared and microwave measurements) and as such…"

P2 L16 There are significant errors associated with rainfall measurements from RADAR. I would not call then 'Highly accurate' (e.g. Krajewski et al (2010) gives a good summary of radar-rainfall uncertainties, focusing on improvements since the key paper by Wilson and Brandes in 1979. A more recent paper which focuses more on the applications in urban hydrology is Thorndahl et al (2017) lists a lot of uncertainties. Attempts to model the uncertainties are given in Villarani et al (2014), Rico-Ramirez et al (2015), Bong-Chul Seo and Krajewski (2015) and Cecinati et al (2017).)
Thank you for the literature on radar uncertainties. The sentence will be modified as follows:
"Radar estimates provide high spatial and temporal resolution estimates of rainfall over local regions, however these estimates can be inaccurate compared to rain gauges (Krajewski et al.

2010, Villarini and Krajewsky 2010 and McKee and Bins 2015), and very few national networks of radar observations exist."

P2 L19 Replace 'global if not quasi-global' with 'global/quasi-global'
The change will be implemented.

P2 P3 I think that GPM should be mentioned as this is the most cutting edge satellite measurement of rainfall, Also mention blended datasets:
MSWEP (https://journals.ametsoc.org/doi/full/10.1175/BAMS-D-17-0138.1), Blended gauge/satellite (https://agupubs.onlinelibrary.wiley.com/doi/pdf/10.1002/2013JD020686) etc.
The following text will be added on P2 L30 before the sentence beginning with "The biggest limitation…":
"New satellite missions and technology will be able to overcome these shortcomings over time. For example, the recently launched Global Precipitation Measurement (GPM) mission is an international satellite mission that aims to improve the detection of light rain and snowfall as well as provide quantitative estimates of precipitation particle size distribution (Hou 2014)."
A note on blended datasets will also be added at the end of the paragraph (P2 L34):
"Very recently, datasets that blend together precipitation estimates from multiple sources such as gauge observations, satellite observation and even reanalyses have become available. Examples include MSWEP V2 (Beck et al. 2019), CHIRPS (Funk et al. 2015) and Shen et al. (2014). These datasets offer very high temporal and spatial resolution data with a reasonably long temporal record. However, these datasets may exhibit increased temporal variability due to the incorporation of various observational sources over time and do not include as many in situ station observations as the gauge-only datasets."

P4 L15 Given that you are taking the GPCC to be the 'best' dataset, I would list that first.
GPCC will be placed first in the list.

P4 L15-18 It would be useful to put in brackets the number of gauges in each data source.
The list will be modified as follows:
"1. Global Precipitation Climatology Centre … (approx. 100,000 stations)
2. Global Historical Climatology Network - Daily … (103,635 stations)
3. Other: Argentina and Russian stations (approx. 1000 stations)"

P4 L18 Include a table of 'other' data sources that can be referred to here. Currently left wondering for a long time what the other data sources are. Is it just Argentina and Russia? If so, just state that.
We will include the information about "other" data as shown in the modified list above.

P4 L24 Re-phrase to be a call to action for met services to share more data.
This series of sentences had to be picked very carefully in order to not offend the agencies that are currently providing GPCC with in situ data. As such we have to be careful not to use strong language, however, in the interest of increasing the sharing of data we can add the following sentence at the end of the paragraph. "We encourage maintainers and providers of data to advocate for increased and more open sharing of meteorological data within their organisations.".

P5 L3 Any idea why there was a decline in 2010?

We suspect that due to the manual data acquisition and quality checks employed by GPCC, recent data is slowly acquired and manually incorporated into the high quality archive.

P6 F2 Awesome! I can't really see any of the 'other' points though
As seen in Figure 1b, there are only 43 unique stations that are incorporated in the end by REGEN. As such they are difficult to spot but they can be spotted in Argentina and Russia (only 3 in Russia).

P6 L7 Clarify if you used existing QC code or if you rewrote it based upon Durre et al.
The following sentence will be added after the first sentence of the paragraph.
"Only minor changes to account for different data formats were made to the original QC procedures from Durre et al. (2010) before applying them."

P7 L1 Describe any validation of the QC code that was undertaken to assess correctly/incorrectly flagged values.
The following sentence can be added at the end of the sentence on P7 L2:
"For a thorough account of the validation of each QC check including the respective false-positive rates, see Durre et al. 2010. The total false positive rate based on all checks is 1% (Durre et al. 2010)."

P7 L11 Why did you choose 70% for a threshold?
The following sentence will be added after the line:
"We chose 70% because the same threshold was used by GPCC for creating their daily gridded products (Schamm et al. 2014). Haylock et al. 2009 (E-Obs) also use a similar threshold of 80%."

P7 L13 Why is there a drop in Indian station data in the 70s?
This is because India has not shared any new data since 1970s. The line can be modified as follows:
"The spike in missing month percentage in South Asia is because there are no Indian stations available after 1970."

P8 L12 'Higly' typo
"Higly" will be changed to Highly

P8 L8 1 degree seems like a huge area to consider gauges to be the same over. Why was such a large area used? How many gauges were merged in this way?
The following sentence will be added at the end of the sentence on P8 L10:
"A search radius of 1 degree was necessary to allow for many stations to be compared with each other in order to account for possible inaccuracies regarding stations metadata (coordinates)."
We understand that this may not be the most intuitive approach. As stated on P8 L11 this approach will be modified in the next iteration of REGEN. Unfortunately, we did not record how many gauges were merged in this way.

P8 L15 Please could you elaborate on the merging process. how were records combined? did you use the whole record from the highest quality source? or did you insert, for example, a few days from a GHCND record into periods of missing data from a GPCC record? If so, did you replace all missing days or did you only replace when there were a whole month of values? Do you have any idea of how many values were merged this way and whether it impacted the homogeneity of records?

No completeness criteria was applied before replacing missing values from higher quality sources. All together there were 36,828 station records that were created by merging records from 2 or more sources.
The following description will be added at the end of the paragraph:
"This way if data from a higher quality source was missing, it was replaced with data from a matching station from a lower quality source but not vice versa. Note that this approach may introduce inhomogeneities in the raw station data."

P8 L29 Was it ever the case that the daily gauges showed no rainfall for the month but the monthly gauge did? If so, what did you do? Also, what do you do with one or more missing days of data? By disaggegating a monthly rainfall value with an incomplete daily record, you will most likely be increasing the average daily rainfall and reported extremes.
We do not calculate ratios for interpolation based on any monthly gauges. The monthly totals are calculated from the daily records. For example, if for a particular month, there are at least 70% of days with non-missing data, then we simply add them up to generate a monthly total. This is explained on P9 L5, however, we can move it to the start of the paragraph after the first sentence.
"The monthly totals for calculating daily ratios in the station timeseries were obtained by summing the daily station data as well. A month was considered complete if it had at least 70% of non-missing days."
As a result, the sentence beginning on P9 L6 will be modified as follows:
"A disadvantage of interpolating anomalies was that even if a daily record existed, it was not used for interpolation if the monthly total was missing because of the completeness criteria."

P8 L30 Please clarify how the monthly data is used. Is it the case that you are effectively temporally disaggregating the monthly rainfall to a daily timestep, ultimately preserving the monthly values? Or will the monthly totals end up being slightly different?
Prior to interpolation the daily ratios (fractions) add up to 1 for each month for each individual station. However, because of the interpolation procedure itself the interpolated grid cell estimates of the ratios may not add up to 1 for the month. As such the monthly totals are not necessarily preserved.

P9 L25 I'm confused by point 3- how are the number of observations different to the number of stations? Is it that you may have 5 stations in the grid box but one of them has no data for that day and so the number of observations is 4? Please clarify.
This is because in some grid boxes with low numbers of stations, stations outside the grid box may be used for interpolation. The last sentence of point 3 will be extended to include this explanation:
"Note that this is not the number of stations used for interpolation of that grid cell estimate, as stations outside the grid cell may be used for interpolation in some cases where station density is low."

P9 L27-35 Please add in some descriptive numbers of how 'wrong' the interpolated rainfall can be.
The sentence starting on P9 L31 will be modified as follows:
"The largest CoV values (maximum of 2.06) are once again seen in Africa, South America, Greenland and Southeast Asia (figures 5c and 5d). This means that the variance between the grid cell estimate and the observations used for interpolation is around twice as large as the average precipitation in these grids. Grids with CoV greater than 1.9 make up less than 0.1% (18 all together) of the grid cells with the mode of CoV being around 1."

P10 L9 Replace 'trends' with 'changes' (we should discourage the use of trends in hydrology: https://doi.org/10.1016/j.advwatres.2017.10.015)

"Trends" will be replaced with "changes" on P10 L9.

P10 L11 replace "we highly encourage uers to" with "Users must"

The sentence will be modified as per the recommendation:

"Users must use a dataset…"

P10 L15-17 This is very unclear. You use 'either' but do not provide an 'or'. Please clarify.

There is an "or" on L16 after "least one station". A comma can be added to make it clearer:

"...either contained at least 60% days in every decade from 1950 to 2013 (7 in total) with at least one station, or both the grid cell coefficient of variation…"

P11 L2 Is this because of the QC applied to GPCC and REGEN?

We do not fully understand why this is the case. We have re-downloaded the data and re-plotted the figure with different climatology periods (see plot below which uses a 1981-2000 climatology instead of 1951-2010 climatology as in the main text) however the enhanced variability remains.

GHCN-Monthly is also QC'd albeit slightly differently to REGEN so we do not think that is the reason behind this variability either. We suspect that this may be due to the coarse resolution (5° x 5° compared to 1° or finer for the other datasets) and poor spatial coverage of GHCN-Monthly dataset (see plot of annual totals for an example year, 1981, using GHCN-Monthly below).

[Figure]

Above: Comparison of annual precipitation anomaly timeseries with monthly datasets. Anomalies were calculated relative to the average of daily precipitation totals over 1981-2000 for each dataset.

Below: Annual total precipitation for 1981 based on GHCN-Monthly V2 gridded precipitation shown as an example of spatial coverage and resolution of the dataset.

[Figure]

P11 L9 Can you provide any estimates of how many more stations are used in national datasets compared to those used in REGEN?

The following table will be included in the main text:

| Regional Dataset Name | Regional Dataset Stations | All REGEN v1.1 stations | Long term (40yr) REGEN v1.1 stations |
|---|---|---|---|
| APHRODITE | Daily max of 8000+ | 8551 | 1539 |
| SA-Obs v1.0 | 7956 | 2527 | 64 |
| E-Obs | 17,468 | 28,338 | 11,261 |
| CPC CONUS | ~28,500 | 42,229 | 3940 |
| AWAP | Daily max of ~7500 | 12,993 | 1424 |

P11 L10 State what interpolation method E-Obs used.
We can add a note in the parenthesis after mentioning E-Obs on P11 L10:
"...and both 0.25 degree and 1 degree versions of E-Obs Version 16 (Haylock et al., 2008; note that E-Obs also uses CAI with global Kriging to interpolate the daily anomalies)
for the events…"

P12 L19 Comma after Africa
A comma will be added after "Africa".

P13 L17 Remove 'running'
Running will be deleted from the sentence.

P13 L28 Replace 'are encouraged to' with 'should'
The sentence will be modified as follows:
"User of REGEN should use the quality mask…"

P14 L25 It is such a shame that REGEN will not be updated.
We just wanted to clarify that REGEN is not an operational product created by a national meteorological agency but instead created during a PhD project. Resource permitting, we would of course continue maintaining REGEN as long as possible.

P14 Include a discussion of the limitations of 1 degree dataset. Rainfall is highly spa- tially variable and a 100km2 estimate is unlikely to contain the information necessary for many typical rainfall applications.
The start of the paragraph on P14 L16 will be modified as follows:
"Rainfall is highly variable and a 1 degree spatial resolution (roughly a 10,000 sq. km) dataset such as REGEN is unlikely to contain the information necessary for many typical local-to-regional rainfall applications. However, we note that actual rainfall amounts in gridded datasets are subject to large uncertainties anyway (ref. Herold et al 2016, GRL), which likely complicated hydrological applications, whereas estimates of variability are more robust. We therefore believe that REGEN will prove itself valuable for climatological applications including studies of climate variability and long-term changes in daily precipitation intensity and extremes, as it provides  long temporal coverage of quasi-global daily precipitation observations."

P14 L31 Why should we expect differences in the total annual precipitation between REGEN and REGEN-40 if they have both been adjusted by the monthly data? Are the monthly totals not necessarily preserved?
As explained with regards to the comment on P8 L30, the monthly totals are not necessarily preserved after interpolation. Since the station distribution between REGEN and REGEN40YR is different, the interpolated fields of ratios will also be different, and hence the monthly totals will in turn be different as well.

P15 L1 Replace 'REGEN has proved itself by providing' with 'REGEN provides'
The sentence will now begin with "REGEN provides"

P15 L4 Remove 'To this note' ad include 'therefore' after are.
The sentence will be modified as follows:
"REGEN and its variant REGEN40YR (which minimises station network variability) are therefore accompanied by various uncertainty estimates as well as a quality mask…"

P15 L7 Include a statement about copyright/useage. Can anyone use this dataset freely? Industry? Or just for research?
The following copyright statement will be added:
"Licence & Rights:
Non-Commercial Licence: CC-BY-NC-SA
Creative Commons - Attribution - Non Commercial - No Derivatives 4.0 International
http://creativecommons.org/licenses/by-nc-sa/4.0/legalcode
Access to this dataset is free, the users are free to download this dataset and share it with others and adapt it as long as they credit the dataset owners, provide a link to the license, and if changes

were made, indicate it clearly and distribute their contributions under the same license as the original, commercial use is not permitted."

**FIGURES**

Please include labels on all of your figures/scale bars. This makes them much easier to interpret than having to refer to the caption.
Every colour bar for all figures will be modified to have a label above it.

F4.c The lower line is missing.
A bottom border will be added to the box around the map in Fig 4c.

F5 Caption: space needed between (KE) and (figures Missing '(figures 5c and 5d)'
from the caption
The figure caption will be modified as follows:
"Figure 5. Kriging error (KE) (figures 5a and 5b), Coefficient of variation (CoV) (figures 5c and 5d) defined by…"

F9 Please label the columns and rows on the diagram, it would make it much easier to interpret. I think 9h and 9g are mixed up in the caption. Also, Rx1day is not defined anywhere in the text or figures.
Besides labels describing the individual maps above the colour bars, labels will be added in the margins describing the columns and rows.
"Rx1Day" will be replaced by "annual maxima" in accordance with the main text (P12 L8). Also a reference to figure 9 will be added at the end of the sentence (P12 L10):
"... (Chen and Xie, 2008; Xie et al., 2007; Chen et al., 2008) and GPCC Full Data Daily V1 (GPCC-FDD1) (figure 9)."

F10 Please label the columns and rows on the diagram, it would make it much easier to interpret.
Similar modification to Fig 9 will be made to Fig 10.

Style points: - I found the italicisation of latin terms like 'in situ' distracting , especially in the abstract. - Why are you using 'in situ measurements' as opposed to 'gauge measurements' as your terminology?
"In situ" and "gauge" are equivalent. There was no particular reason behind choosing to use "in situ". We will continue to use "in situ" but all italicisation of "in situ" will be removed from the text.

---

## Editor Comment (EC1) · Dimitri Solomatine (Editor) · 19 Jun 2019

The referees were quite positive, but brought to the attention of authors interesting and important points, some of which are absolutely vital for accurate presentation of the rainfall data. The authors in their replies have clearly shown that they understand the comments well, and explained how the manuscript will be updated to address the comments. The authors will be invited to revise the manuscript.

---

## Author Response (AR1)

Dear Editor,

We apologise for the delay in uploading the revised manuscript. We have implemented all changes outlined in the original responses to anonymous referees. In addition to these changes, we have also made changes in accordance with newer versions of REGEN and the long-term REGEN datasets. The newer REGEN datasets (V1-2019) represent an incremental change from the REGEN V1.0 datasets in that none of the original methodologies described in the original manuscript have been changed. Instead, we have simply just extended the temporal record of the datasets by 3 years (1950 - 2016 vs 1950 - 2013). This came about because in late 2018, GPCC released an extended version of their monthly dataset (GPCC Full Data Monthly V2018) which is used by REGEN to facilitate climatologically aided interpolation. For the most part these changes include changes to the period of REGEN from 1950 - 2013 to 1950 - 2016 and changes to the version number of REGEN in text. No changes needed to be made to the description of results or conclusions. Finally, some minor language changes have also been made for better readability without changing the original meaning.

Entirely new figures or figures with any major changes include figures 7a, 7c, 8 and A1, in accordance with the feedback from anonymous referees. Besides these figures, figures 5, 6, 7b, 7d, and 9 - 13 have also been updated because of the new underlying REGEN and REGEN40YR datasets. As requested by referee #2 a new table comparing stations used for interpolation between REGEN and regional datasets has also been added.

Please find a marked up version of the manuscript that highlights all new additions or changes to text accompanied with this response document, in addition to the final manuscript.

We thank both anonymous referees for the insightful and detailed comments that helped improve the manuscript. The text and analysis added as a result certainly helps to clarify any misleading or unclear information and provides more pertinent information about the REGEN datasets.

Below we address each Reviewer's comments individually in red. Page and line numbers in the responses refer to the updated manuscript.

Many thanks,

Steefan Contractor

**Reviewer 1**

*The paper presents and documents a global daily precipitation dataset at a 1 degree resolution. The dataset is compiled from several data sources. In the paper, discussions on the method to grid daily precipitation and the density of the network, including the consequences this has for the precipitation estimates on the grid, are included. In addition to REGEN, a dataset based only on the long-running stations is presented which will be temporally more homogeneous than the dataset based on all station data.*
*The study is a very welcome contribution to the field where for daily precipitation now many national and several regional datasets exist, but (up to now) not a global dataset based on in-situ measurements. The study is well written and clear and as far as I can judge, no problems in the analysis are there. However, the study could do with a more expansive and in-depth comparison*

*against existing (regional) observational datasets - the current comparisons are too ad-hoc and uninformative.*

*My advise to the editor is to accept the paper with minor adjustments.*

We thank the reviewer for their thoughtful and thoroughly researched comments and agree with the major criticism about the lack of a more in-depth comparison with regional datasets.

**More Serious Concerns**

*1. Metadata is often a problem (lacking, erroneous etc.). The day shift discussed in sect. 2.3 - a very necessary thing to do - is in the face of poor metadata an action which might be problematic. The give an example, according to the Appendix, the data from the Netherlands are shifted one day backward in time. This is appropriate for the manual rain gauges but not for the 30+ automatic weather stations which measure precip between 0-0 UTC. It could be that only the rain gauges are in the GPCC dataset - but the reader can't tell.*

*Checking with other NMHSs in Europe, I could confirm the necessity to shift the date, except for Hungary (there is a question to the Hungarian NMHS out now).*

*There is another confusing part of this date-adjustment. Are you aiming to get 24-hour values coinciding with a day defined by local time or by UTC? The reason for asking is that for Indonesia, my understanding is that measurements are from 7 - 7 local time, which is (nearly) O - O UTC.*

*The correlations with CPC (figure 11d) are low in some areas - could it be that an erroneous timeshift or a missed time shift could be related to the low correlation? I guess you have tried to shift the whole dataset back and forth and looked for areas on the globe with increases in correlation?*

We agree with the reviewer about the problems associated with poor metadata and the day shift discussed in the text. We are aware of automated weather stations that measure precipitation over a different 24h window to the country they are in. In the text we have identified a similar issue with 10% of American stations which were also automatic weather stations (P8 L19). We have expanded the note on this issue as follows:

"Note that some countries maintain a mix of manually monitored and automated weather stations which may represent precipitation over differing 24h windows that may not be suitable for being shifted identically. For example, around 10% of observations in the US and around 30 stations in the Netherlands are midnight observations, i.e. observations over the 24h period from midnight to midnight UTC which are assigned to the day on which the observing period ends. Although these observations have not been manually adjusted in this version of REGEN, they will be taken care of in the next iteration. Globally more countries may exist whose gauge observations may represent a mix of reporting times (due to the use of automatic weather stations for example), however, without proper metadata about these reporting times it is not possible for us to adjust their timestamp accordingly."

Regarding the hungarian data, it in fact has not been shifted by us. The inclusion of "Hungary" in the list of shifted stations in the appendix was an error. It has thus been removed from the list. We thank the reviewer for bringing this to our attention.

Dates are not adjusted to match 0-0 UTC time but rather to match the local 0700h-0700h local time to preserve the diurnal aspect of precipitation.

It is possible that correlations between REGEN and CPC are lower because of this shifting. Based on the correlation between REGEN V1.1 All Stations shifted +1 days and CPC (shown below) we see that the correlations are higher in and around the countries where data was shifted a day back

(eg. Vietnam, Brazil, Uruguay, Peru, Suriname, Netherlands, Norway, Ukraine and Turkey). Lower correlations are observed in all other regions.

[Figure]

REGEN V1.1 lagged +1 days vs CPC_Glb temporal correlation 1988-2013

The above figure has been added to the appendix (Figure A1) and the following note will be added to the text on P15 L12.

"Correlations between REGEN and CPC may be lower in parts where the underlying stations were shifted a day backward (see Appendix). Indeed, based on correlations between REGEN lagged +1 days and CPC (Fig. A1), the correlations are higher compared to figure 11d in and around the countries where data was shifted a day back (eg. Vietnam, Brazil, Uruguay, Peru, Suriname, Netherlands, Norway, Ukraine and Turkey). Correlations do not change compared to figure 12d in all regions where REGEN raw station data are not shifted."

*2.	The comparison against regional dataset of daily precipitation (sect. 3.2) is too ad-hoc. In your article, you claim (rightly so) that national and regional datasets are based on a more extensive dataset. I would like to add that especially national datasets have a far greater detail in the understanding of the metadata. This means that a meaningful comparison can be made between national/regional datasets and the REGEN dataset.  This should go beyond simply picking one event of a few days, averaging precip over a region and plot a few timeseries.*

*Please add a more expansive comparison with regional datasets like Aphrodite etc. Other datasets might be interesting to use as well, like the SA-OBS for Southeast Asia (van den Besselaar et al. 2017. doi:10.1175/JCLI-D-16-0575.1). Comparison you could make easily are the standard deviation of the daily difference, perhaps stratified over different periods, but other comparison metrics are equally useful. Given the particular focus on precipitation extremes by the international community - and of some of the authors of the paper - a dedicated focus of representation of extremes (beyond one example) is required.*

*I was taken by surprise when reading about the the "Great flood of 1968" in Southeast England and France. The article referred to (Jackson, 1977) never mentions France. Below are a few pictures*

*from the E-OBSv19.0e for 14-16 September 1968 and the area you use to compare REGEN with the regional dataset is somewhat large compared to the area affected by this event.*

We have added a new figure (figure 8) that compares the mean and standard deviation of difference between REGEN and the regional datasets in the main text. These include those datasets already mentioned in the paper (CPC CONUS, E-Obs, Aphrodite, AWAP) as well as the SA-Obs dataset highlighted by the reviewer. In addition to the above comparison, figure 8 also shows maps of temporal correlations of local (grid-cell) timeseries between the regional datasets and REGEN to show a comparision of temporal variability between the dataset pairs. The results of the comparison have been discussed on P13 L1-23. Finally, as suggested by anonymous referee #2, we have also included a table (table 1) comparing number of stations between the regional datasets, REGEN and REGEN40YR. This table is shown below.

| Regional Dataset Name | Regional Dataset Stations | All REGEN v1.1 stations | Long term (40yr) REGEN v1.1 stations |
|---|---|---|---|
| APHRODITE | Daily max of 8000+ | 8551 | 1539 |
| SA-Obs v1.0 | 7956 | 2527 | 64 |
| E-Obs | 17,468 | 28,338 | 11,261 |
| CPC CONUS | ~28,500 | 42,229 | 3940 |
| AWAP | Daily max of ~7500 | 12,993 | 1424 |

We will modify figure 7a so only UK is included in the spatial averaging and any mention of France can be removed. The updated figure is shown below. In addition, figure 7c now includes an additional timeseries based on SA-Obs V1.

[Figure]

Other issues the authors may want to look into

*1.      page 2, line 16: here is is claimed that radar provides 'highly accurate' estimates of precipitation.  It is my understanding that radar   can underestimate extreme precipitation by as much as 40% (e.g. https://journals.ametsoc.org/doi/10.1175/2009BAMS2747.1)*

The sentence has been replaced with the following on P2 L18:
"Radar estimates provide high spatial and temporal resolution estimates of rainfall over local regions, however these estimates can be inaccurate compared to rain gauges (Krajewski et al. 2010 Villarini and Krajewsky 2010 and McKee and Bins 2015), and very few national networks of radar observations exist.".

*2.      the referencing to figures is a bit curious: The first 3 are referenced chronologically, but then on page 7, you refer to fig. 4b, the next reference (line 31) is to fig. 7b page 9 10 have refs to fig 5 and page 11 has a reference to fig 7.*

Besides the reference to Figure 7b all other references are in order of figure appearance (see locations below). Figure 7b is referenced earlier as it helps to demonstrate the time-shifting of data. We leave it to the editor to clarify whether chronological figure referencing is strictly necessary, or whether referencing the Figure 7 once a little earlier is acceptable to improve readability.
Fig 4a,b: P6 L12, P7 L5
Fig 5a-f: P9 L26-27, P10 L10
Fig 6: P10 L25

*3.     page 8, lines 4-7. Relocated stations often keep the WMO id and if the relocation is to a site in the vicinity, then your criterion labels the old and new station as the same. This may not be a problem for precipitation, but perhaps it is good to inform the reader about this.*

The following note has been added to the text on P9 L5:
"Also note that WMO station IDs do not change after a station is relocated to a site in the vicinity which can result in two stations in different locations merged together according to our criteria."

*4.     page 11, line 10. There is no 1.0 degree version of the E-OBS. I guess you regridded the E-OBS data to the REGEN grid to arrive at the 1 degree resolution?*

The 1 deg version of E-Obs was created by regridding the 0.25 deg product using second order conservative remapping from CDO (cdo remapcon2). P12 L14 has been updated as follows to reflect this.
"There is good agreement between the daily timeseries from REGEN, REGEN40YR and both 0.25 degree and 1 degree (*regridded from 0.25 degree version using CDO remapcon2)* versions of E-Obs Version 16".
The figure label has also been updated (see figure above).

*5.     page 14, line 19-20. Here you make the point that there is an ordering in the number of stations used by national, regional and global datasets. The point is very valid, but the example provided is misleading. Herrera used 2756 stations, the E-OBS uses 210 station in Spain (incl. Catalonia) and for the whole of Europe, 15962 series are used. Hardly 'roughly the same number' as claimed.*

The sentence is false. We meant to say that the Spanish Meteorological Agency (AEMET) maintains roughly the same number of stations as those used in the entirety of Europe by E-Obs. The text (P17 L12) has been modified to reflect this as follows:
"For example, the Spanish Meteorological Agency (AEMET) itself manages roughly over 9000 stations (Hererra et. al. 2012) which is almost the same number of stations as those used by E-Obs for the entirety of Europe (around 12,000 gauges at its maximum)."

**Very minor issues**

*page 3, line 4, It is the Climatic Research Unit (not Climate)*
"Climate" has been changed to "Climatic" on P3 L13.

*page 7, line 4, typo in procedures*
"prodcedures" has been changed to "procedures" on P7 L23. Another typographical error on the same line has also been fixed.

*page 9, line 20, perhaps an odd formulation?*
The sentence has been modified as follows on P10 L18:
"Kriging error. This is not an absolute error but rather…"

*references, many citations have the http address twice in the citation, e.g. Jackson (1977).*

The repeated links have been removed from the following citations:
Yamamoto 2000, Xie et al 2007, Tian, Y. and Peters-Lidard 2010, Smith et al 2010, Schneider et al 2014, Peterson et al 1997, Perry and Hollis 2005, Osborne and Hulme 1998, Jackson 1977, Isotta et al 2013, Hofstra et al 2008, Herrera et al 2012, Harris et al 2014, Groisman et al 2005, Funk et al 2015, Frei et al 1998, Donat et al 2013b, Chen et al 2008, Bytheway and Kummerow 2013, Ashouri et al 2014, Allen and Ingram 2002, Alexander et al 2006, and Adler et al 2003.

*Appendix A. you apparently made an effort to make an alphabetical list - but didn't quite succeed. There are duplicates in the list too - like Indonesia.*
The countries have been reordered in alphabetical order. Also repeat entries of Georgia and Indonesia have been removed, and "Guam" has been removed from this list as it is a US territory.

*caption fig. 5: fig. c d show what?*
The figure caption has been modified as follows:
"Figure 5. Kriging error (KE) (figures 5a and 5b), Coefficient of variation (CoV) (figures 5c and 5d) defined by…"

**Reviewer 2**

*This paper presents a new global gridded rainfall dataset that combines existing gauge datasets, quality controls the data and produces a useful new gridded daily rainfall product. Rainfall data at a daily timestep are difficult to access and in parts of the world, not even collected. The work presented here represents a substantial effort and contribution to both the literature on rainfall data and in providing a new resource for hydrologists amongst others. Well done!*

*The paper is well written and the methods and analysis are sound. I think the paper only requires some minor revisions to be acceptable for publishing.*

We thank the reviewer for the encouraging review. The suggestions and points raised help clarify any misleading information and provide additional detailed documentation of pertinent information related to REGEN.

*P2 L9: ALL measurements of PPT have errors including gauges. Please clarify this and add in a short discussion of the errors in gauge data. (see McMillan et al 2012 as a starting point)*

The sentence on P2 L8 has been replaced with the following paragraph.
"All observations have errors, for example, gauge-based precipitation measurements are subject to undercatch, wind related errors, evaporation loss, wetting loss, splash in/out errors and tipping errors (see McMillan et al 2012 for details). However, alternatives to gauge-based measurements such as satellite observations, model reanalysis products and radar-based observations have additional limitations."

*P2 L10: Who thinks that reanalysis products represent the 'true state of the system'? Either cite or say that reanalysis data are sometimes misused as observations*

The sentence has been modified as follows on P2 L12:
"They are often misused as observations but in fact inherit…"

*P2 L13 NASA MERRA2 assimilates precipitation*
*(https://gmao.gsfc.nasa.gov/pubs/docs/McCarty885.pdf Table 1)*

The sentence has been modified as follows on P2 L13:

"Furthermore, none of the reanalysis products assimilate gauge-based precipitation observations (MERRA2 however incorporates satellite infrared and microwave measurements) and as such…"

*P2 L16 There are significant errors associated with rainfall measurements from RADAR. I would not call then 'Highly accurate' (e.g. Krajewski et al (2010) gives a good summary of radar-rainfall uncertainties, focusing on improvements since the key paper by Wilson and Brandes in 1979. A more recent paper which focuses more on the applications in urban hydrology is Thorndahl et al (2017) lists a lot of uncertainties. Attempts to model the uncertainties are given in Villarani et al (2014), Rico-Ramirez et al (2015), Bong-Chul Seo and Krajewski (2015) and Cecinati et al (2017).)*

Thank you for the literature on radar uncertainties. The sentence has been modified as follows on P2 L18:

"Radar estimates provide high spatial and temporal resolution estimates of rainfall over local regions, however these estimates can be inaccurate compared to rain gauges (Krajewski et al. 2010, Villarini and Krajewsky 2010 and McKee and Bins 2015), and very few national networks of radar observations exist."

*P2 L19 Replace 'global if not quasi-global' with 'global/quasi-global'*
The change has been implemented on P2 L21.

*P2 P3 I think that GPM should be mentioned as this is the most cutting edge satellite measurement of rainfall,     Also mention blended datasets:*
*MSWEP (https://journals.ametsoc.org/doi/full/10.1175/BAMS-D-17-0138.1),        Blended gauge/satellite (https://agupubs.onlinelibrary.wiley.com/doi/pdf/10.1002/2013JD020686) etc.*
The following text has been added on P2 L32 before the sentence beginning with "The biggest limitation…":

"New satellite missions and technology will be able to overcome these shortcomings over time. For example, the recently launched Global Precipitation Measurement (GPM) mission is an international satellite mission that aims to improve the detection of light rain and snowfall as well as provide quantitative estimates of precipitation particle size distribution (Hou 2014)."

A note on blended datasets has also been added at the end of the paragraph (P3 L4):

"Very recently, datasets that blend together precipitation estimates from multiple sources such as gauge observations, satellite observation and even reanalyses have become available. Examples include MSWEP V2 (Beck et al. 2019), CHIRPS (Funk et al. 2015) and Shen et al. (2014). These datasets offer very high temporal and spatial resolution data with a reasonably long temporal record. However, these datasets may exhibit increased temporal variability due to the incorporation of various observational sources over time and do not include as many in situ station observations as the gauge-only datasets."

*P4 L15 Given that you are taking the GPCC to be the 'best' dataset, I would list that first.*
GPCC has been placed first in the list.

*P4 L15-18 It would be useful to put in brackets the number of gauges in each data source.*
The list has been modified as follows on P4 L23-27:
"1. Global Precipitation Climatology Centre … (approximately 100,000 stations)
2. Global Historical Climatology Network - Daily … (103,635 stations)
3. Other: Argentina and Russian stations (approximately 1000 stations)"

*P4 L18 Include a table of 'other' data sources that can be referred to here. Currently left wondering for a long time what the other data sources are. Is it just Argentina and Russia? If so, just state that.*

We have included the information about "other" data as shown in the modified list above.

*P4 L24 Re-phrase to be a call to action for met services to share more data.*

This series of sentences had to be picked very carefully in order to not offend the agencies that are currently providing GPCC with in situ data. As such we have to be careful not to use strong language, however, in the interest of increasing the sharing of data we have added the following sentence at the end of the paragraph (P5 L8):

"We encourage maintainers and providers of data to advocate for increased and more open sharing of meteorological data within their organisations.".

*P5 L3 Any idea why there was a decline in 2010?*

We suspect that due to the manual data acquisition and quality checks employed by GPCC, recent data is slowly acquired and manually incorporated into the high quality archive.

*P6 F2 Awesome! I can't really see any of the 'other' points though*

As seen in Figure 1b, there are only 43 unique stations that are incorporated in the end by REGEN. As such they are difficult to spot but they can be spotted in Argentina and Russia (only 3 in Russia).

*P6 L7 Clarify if you used existing QC code or if you rewrote it based upon Durre et al.*

The following sentence has been added after the first sentence of the paragraph (P7 L6).

"Only minor changes to account for different data formats were made to the original QC procedures from Durre et al. (2010) before applying them."

*P7 L1 Describe any validation of the QC code that was undertaken to assess correctly/incorrectly flagged values.*

The following sentence can be added at the end of the sentence on P7 L1:

"For a thorough account of the validation of each QC check including the respective false-positive rates, see Durre et al. 2010. The total false positive rate based on all checks is 1% (Durre et al. 2010)."

*P7 L11 Why did you choose 70% for a threshold?*

The following sentence will be added after P7 L27:

"We chose 70% because the same threshold was used by GPCC for creating their daily gridded products (Schamm et al. 2014). Haylock et al. 2009 (E-Obs) also use a similar threshold of 80%."

*P7 L13 Why is there a drop in Indian station data in the 70s?*

This is because India has not shared any new data since 1970s. The line can be modified as follows on P7 L29:

"The spike in missing month percentage in South Asia is because there are no Indian stations available after 1970."

*P8 L12 'Higly' typo*

"Higly" will be changed to Highly on P8 L33.

*P8 L8 1 degree seems like a huge area to consider gauges to be the same over. Why was such a large area used? How many gauges were merged in this way?*

The following sentence has been added at the end of the sentence on P8 L30:

"A search radius of 1 degree was necessary to allow for many stations to be compared with each other in order to account for possible inaccuracies regarding stations metadata (coordinates)."

We understand that this may not be the most intuitive approach. As stated on P8 L34 this approach will be modified in the next iteration of REGEN. Unfortunately, we did not record how many gauges were merged in this way.

*P8 L15 Please could you elaborate on the merging process. how were records combined? did you use the whole record from the highest quality source? or did you insert, for example, a few days from a GHCND record into periods of missing data from a GPCC record? If so, did you replace all missing days or did you only replace when there were a whole month of values? Do you have any idea of how many values were merged this way and whether it impacted the homogeneity of records?*

No completeness criteria was applied before replacing missing values from higher quality sources. All together there were 36,828 station records that were created by merging records from 2 or more sources.

The following description has been added at the end of the paragraph on P9 L5:

"This way if data from a higher quality source was missing, it was replaced with data from a matching station from a lower quality source but not vice versa. Note that this approach may introduce inhomogeneities in the raw station data."

*P8 L29 Was it ever the case that the daily gauges showed no rainfall for the month but the monthly gauge did? If so, what did you do? Also, what do you do with one or more missing days of data? By disaggegating a monthly rainfall value with an incomplete daily record, you will most likely be increasing the average daily rainfall and reported extremes.*

We do not calculate ratios for interpolation based on any monthly gauges. The monthly totals are calculated from the daily records. For example, if for a particular month, there are at least 70% of days with non-missing data, then we simply add them up to generate a monthly total. This is explained on P9 L21, however, we can move it to the start of the paragraph after the first sentence.

"The monthly totals for calculating daily ratios in the station timeseries were obtained by summing the daily station data as well. A month was considered complete if it had at least 70% of non-missing days."

As a result, the sentence beginning on P9 L33 will be modified as follows:

"A disadvantage of interpolating anomalies was that even if a daily record existed, it was not used for interpolation if the monthly total was missing because of the completeness criteria."

*P8 L30 Please clarify how the monthly data is used. Is it the case that you are effectively temporally disaggregating the monthly rainfall to a daily timestep, ultimately preserving the monthly values? Or will the monthly totals end up being slightly different?*

Prior to interpolation the daily ratios (fractions) add up to 1 for each month for each individual station. However, because of the interpolation procedure itself the interpolated grid cell estimates of the ratios may not add up to 1 for the month. As such the monthly totals are not necessarily preserved.

*P9 L25 I'm confused by point 3- how are the number of observations different to the number of stations? Is it that you may have 5 stations in the grid box but one of them has no data for that day and so the number of observations is 4? Please clarify.*

This is because in some grid boxes with low numbers of stations, stations outside the grid box may be used for interpolation. The last sentence of point 3 has been extended to include this explanation on P10 L19:

"Note that this is not the number of stations used for interpolation of that grid cell estimate, as stations outside the grid cell may be used for interpolation in some cases where station density is low."

*P9 L27-35 Please add in some descriptive numbers of how 'wrong' the interpolated rainfall can be.*

The sentence starting on P10 L25 has been modified as follows:

"The largest CoV values (maximum of 2.33) are once again seen in Africa, South America, Greenland and Southeast Asia (figures 5c and 5d). This means that the variance between the grid cell estimate and the observations used for interpolation is around twice as large as the average precipitation in these grids. Grids with CoV greater than 1.9 make up less than 0.05% (22 all together) of the grid cells with the mode of CoV being around 1.25."

*P10 L9 Replace 'trends' with 'changes' (we should discourage the use of trends in hydrology: https://doi.org/10.1016/j.advwatres.2017.10.015)*

"Trends" has been replaced with "changes" on P11 L7.

*P10 L11 replace "we highly encourage uers to" with "Users must"*

The sentence will be modified as per the recommendation on P11 L9:

"Users must use a dataset…"

*P10 L15-17 This is very unclear. You use 'either' but do not provide an 'or'. Please clarify.*

There is an "or" on L16 after "least one station". A comma can be added to make it clearer on P11 L14:

"...either contained at least 60% days in every decade from 1950 to 2013 (7 in total) with at least one station, or both the grid cell coefficient of variation…"

*P11 L2 Is this because of the QC applied to GPCC and REGEN?*

We do not fully understand why this is the case. We have re-downloaded the data and re-plotted the figure with different climatology periods (see plot below which uses a 1981-2000 climatology instead of 1951-2010 climatology as in the main text) however the enhanced variability remains. GHCN-Monthly is also QC'd albeit slightly differently to REGEN so we do not think that is the reason behind this variability either. We suspect that this may be due to the coarse resolution (5° x 5° compared to 1° or finer for the other datasets) and poor spatial coverage of GHCN-Monthly dataset (see plot of annual totals for an example year, 1981, using GHCN-Monthly below).

[Figure]

Above: Comparison of annual precipitation anomaly timeseries with monthly datasets. Anomalies were calculated relative to the average of daily precipitation totals over 1981-2000 for each dataset.

Below: Annual total precipitation for 1981 based on GHCN-Monthly V2 gridded precipitation shown as an example of spatial coverage and resolution of the dataset.

[Figure]

*P11 L9 Can you provide any estimates of how many more stations are used in national datasets compared to those used in REGEN?*

The following table has been included in the main text:

| Regional Dataset Name | Regional Dataset Stations | All REGEN v1.1 stations | Long term (40yr) REGEN v1.1 stations |
|---|---|---|---|
| APHRODITE | Daily max of 8000+ | 8551 (daily max of 4985) | 1539 (daily max of 1743) |
| SA-Obs v1.0 | 7956 | 2527 | 64 |
| E-Obs | 17,468 | 28,338 | 11,261 |
| CPC CONUS | ~28,500 | 42,229 | 3940 |
| AWAP | Daily max of ~7500 | 12,993 (daily max of 7509) | daily max of 3909 |

*P11 L10 State what interpolation method E-Obs used.*

We have added a note in the parenthesis after mentioning E-Obs on P12 L10:

"...and both 0.25 degree and 1 degree versions of E-Obs Version 16 (Haylock et al., 2008; note that E-Obs also uses CAI with global Kriging to interpolate the daily anomalies)

for the events…"

*P12 L19 Comma after Africa*

A comma will be added after "Africa" on P14 L31.

*P13 L17 Remove 'running'*
Running has been deleted from the sentence on P16 L1.

*P13 L28 Replace 'are encouraged to' with 'should'*
The sentence has been modified as follows on P16 L12:
"User of REGEN should use the quality mask…"

*P14 L25 It is such a shame that REGEN will not be updated.*
We just wanted to clarify that REGEN is not an operational product created by a national meteorological agency but instead created during a PhD project. Resource permitting, we would of course continue maintaining REGEN as long as possible.

*P14 Include a discussion of the limitations of 1 degree dataset. Rainfall is highly spa- tially variable and a 100km2 estimate is unlikely to contain the information necessary for many typical rainfall applications.*
The start of the paragraph on P16 L33 has been modified as follows:
"Rainfall is highly variable and a 1 degree spatial resolution (roughly a 10,000 sq. km) dataset such as REGEN is unlikely to contain the information necessary for many typical local-to-regional rainfall applications. However, we note that actual rainfall amounts in gridded datasets are subject to large uncertainties anyway (ref. Herold et al 2016, GRL), whereas estimates of variability are more robust. We therefore believe that REGEN will prove itself valuable for climatological applications including studies of climate variability and long-term changes in daily precipitation intensity and extremes, as it provides  long temporal coverage of quasi-global daily precipitation observations."

*P14 L31 Why should we expect differences in the total annual precipitation between REGEN and REGEN-40 if they have both been adjusted by the monthly data? Are the monthly totals not necessarily preserved?*
As explained with regards to the comment on P8 L30, the monthly totals are not necessarily preserved after interpolation. Since the station distribution between REGEN and REGEN40YR is different, the interpolated fields of ratios will also be different, and hence the monthly totals will in turn be different as well.

*P15 L1 Replace 'REGEN has proved itself by providing' with 'REGEN provides'*
The sentence will now begin with "REGEN provides" on P17 L23.

*P15 L4 Remove 'To this note' ad include 'therefore' after are.*
The sentence has been modified as follows on P17 L26:
"REGEN and its variant REGEN40YR (which minimises station network variability) are therefore accompanied by various uncertainty estimates as well as a quality mask…"

*P15 L7 Include a statement about copyright/useage. Can anyone use this dataset freely? Industry? Or just for research?*
The following copyright statement has been added on P18 L4:
"Licence & Rights:
Non-Commercial Licence: CC-BY-NC-SA
Creative Commons - Attribution - Non Commercial - No Derivatives 4.0 International
http://creativecommons.org/licenses/by-nc-sa/4.0/legalcode

Access to this dataset is free, the users are free to download this dataset and share it with others and adapt it as long as they credit the dataset owners, provide a link to the license, and if changes were made, indicate it clearly and distribute their contributions under the same license as the original, commercial use is not permitted."

**FIGURES**

*Please include labels on all of your figures/scale bars. This makes them much easier to interpret than having to refer to the caption.*
We found labels above all colour bars made the figures cluttered. We therefore opted for column and row labels in the margins of all multi-map figures (figs 3, 5, and 8-13).

*F4.c The lower line is missing.*
A bottom border has been added to the box around the map in Fig 4c.

*F5 Caption: space needed between (KE) and (figures Missing '(figures 5c and 5d)'*
*from the caption*
The figure caption has been modified as follows:
"Figure 5. Kriging error (KE) (figures 5a and 5b), Coefficient of variation (CoV) (figures 5c and 5d) defined by…"

*F9 Please label the columns and rows on the diagram, it would make it much easier to interpret. I think 9h and 9g are mixed up in the caption. Also, Rx1day is not defined anywhere in the text or figures.*
Labels have been added in the margins describing the columns and rows.
"Rx1Day" will be replaced by "annual maxima" in accordance with the main text (P14 L20). Also a reference to figure 10 (originally figure 9) will be added at the end of the sentence (P14 L22):
"... (Chen and Xie, 2008; Xie et al., 2007; Chen et al., 2008) and GPCC Full Data Daily V1 (GPCC-FDD1) (figure 9)."

*F10 Please label the columns and rows on the diagram, it would make it much easier to interpret.*
Similar modification to Fig 10 (9) have been made to Fig 11, and also to Figs 3, 5, and 8-13.

*Style points: - I found the italicisation of latin terms like 'in situ' distracting , especially in the abstract. - Why are you using 'in situ measurements' as opposed to 'gauge measurements' as your terminology?*
"In situ" and "gauge" are equivalent. There was no particular reason behind choosing to use "in situ". We will continue to use "in situ" but all italicisation of "in situ" will be removed from the text.

**Rainfall Estimates on a Gridded Network (REGEN) - A global land-based gridded dataset of daily precipitation from 1950–2016**

Steefan Contractor[1,2], Markus G. Donat[1,3,4], Lisa V. Alexander[1,3], Markus Ziese[5], Anja Meyer-Christoffer[5], Udo Schneider[5], Elke Rustemeier[5], Andreas Becker[5], Imke Durre[6], and Russell S. Vose[6]

[1]Climate Change Research Centre, UNSW Sydney, Australia
[2]ARC Centre of Excellence for Climate System Science
[3]ARC Centre of Excellence for Climate Extremes
[4]Barcelona Supercomputing Center, Barcelona, Spain
[5]Global Precipitation Climatology Centre, Deutscher Wetterdienst, Offenbach Germany
[6]National Centers for Environmental Information, National Oceanic and Atmospheric Administration, Asheville NC, USA

**Correspondence:** Steefan Contractor (s.contractor@unsw.edu.au)

**Abstract.** We present a new global land-based daily precipitation dataset from 1950 using an interpolated network of in situ data called **R**ainfall **E**stimates on a **G**ridd**E**d **N**etwork - **REGEN**. We merged multiple archives of in situ data including two of the largest archives, the Global Historical Climatology Network - Daily (GHCN-Daily) hosted by National Centres of Environmental Information (NCEI), USA and one hosted by the Global Precipitation Climatology Centre (GPCC) operated by Deutscher Wetterdienst (DWD). This resulted in an unprecedented station density compared to existing datasets. The station timeseries were quality controlled using strict criteria and flagged values were removed. Remaining values were interpolated to create area average estimates of daily precipitation for global land areas on a $1° \times 1°$ latitude-longitude resolution. Besides the daily precipitation amounts, fields of standard deviation, Kriging error and number of stations are also provided. We also provide a quality mask based on these uncertainty measures. For those interested in a dataset with lower station network variability we also provide a related dataset based on a network of long-term stations which interpolates stations with a record length of at least 40 years. The REGEN datasets are expected to contribute to the advancement of hydrological science and practice by facilitating studies aiming to understand changes and variability in several aspects of daily precipitation distributions, extremes, and measures of hydrological intensity. Here we document the development of the dataset and guidelines for best practices for users with regards to the two datasets.

15 *Copyright statement.*

**1 Introduction**

Earth's climate is changing leading to spatial and temporal variations in precipitation. These changes in precipitation are strongly linked to social, economic and environmental prosperity due to the role precipitation plays in global food production

and maintaining biodiversity. Theoretical expectations are that the global hydrological cycle would intensify in a warmer climate, associated with increases in mean and extreme precipitation (whereby mean/total precipitation would increase at lower rate than extreme precipitation due to energetic constraints (Allen and Ingram, 2002)). In addition to changes in precipitation due to climate change, precipitation is also characterised by strong variability in many regions. Reliable observations are necessary to understand these short- and long-term changes and to evaluate climate models which help understand the processes driving these changes. Hence in some ways gridded observations of the past also help us to better plan for and adapt to these changes in the future.

All observations have errors, for example, gauge-based precipitation measurements are subject to undercatch, wind related errors, evaporation loss, wetting loss, splash in/out errors and tipping errors (see McMillan et al. (2012) for details). However, alternatives to gauge-based measurements such as satellite observations, model reanalysis products and radar-based observations have additional limitations. Reanalysis products assimilate observations and models to create a synthesised estimate of the state of the earth system. They are often misused as observations but in fact inherit issues from the incomplete observations and imperfect models and are based on complex assimilation techniques. Furthermore, none of the reanalysis products assimilate surface precipitation observations (MERRA2 however incorporates satellite infrared and microwave measurements) and as such are not representative of reality. This is evidenced by the classification of precipitation as the least reliable class by Kalnay et al. (1996). Renalyses also contain temporal inhomogeneities due to the changing amount of assimilated observations over time (Compo et al., 2006). According to Lorenz and Kunstmann (2012) even the state-of-the-art reanalyses are unsuitable for climate trend and long-term water budget analysis. Radar estimates provide high spatial and temporal resolution estimates of rainfall over local regions, however these estimates can be inaccurate compared to rain gauges (Krajewski et al., 2010; Villarini and Krajewski, 2010; McKee and Binns, 2016), and very few national networks of radar observations exist.

Satellite products have become available in recent years. These datasets are gridded and boast a global/quasi-global coverage. The Tropical Rainfall Measuring Mission (TRMM) 3B42 (Huffman et al., 2007), Global Precipitation Climatology Projects 1 Degree Daily (GPCP-1DD) (Huffman et al., 2001), Climate Hazards Group InfraRed Precipitation with Stations (CHIRPS) (Funk et al., 2015) and the Precipitation Estimates from Remotely Sensed Information using Artificial Neural Networks - Climate Data Record (PERSIANN-CDR) (Ashouri et al., 2014) are some examples of popular satellite based precipitation products. These satellite based datasets, however, use complex algorithms to derive precipitation estimates from indirect radiation measurements resulting in large uncertainties in precipitation estimates. For example GPCP-1DD measures infrared reflectivity of clouds to infer the cloud thickness and then estimates precipitation rates based on the poor relationship between clouds and rainfall (Kidd and Levizzani, 2011). This estimate is also adjusted based on monthly gauge observations, however, the uncertainties remain high. In general satellite products perform well in the tropics where the rain rates are higher but struggle with snow and ice and on complex terrain (Bytheway and Kummerow, 2013; Tian and Peters-Lidard, 2010; Contractor et al., 2015). New satellite missions and technology will be able to overcome these shortcomings over time. For example, the recently launched Global Precipitation Measurement (GPM) mission is an international satellite mission that aims to improve the detection of light rain and snowfall as well as provide quantitative estimates of precipitation particle size distribution (Hou et al., 2014). The biggest limitation of satellite products, however, is also their brevity. It was only after Tropical Rainfall

Measurement Mission (TRMM) in 1997 where we entered an era of multi-sensor measurements across multiple satellites to produce a globally consistent and complete map of precipitation (Tian and Peters-Lidard, 2010). Thus the satellite products do not allow for an analysis of global rainfall changes that effectively separates the natural variability from anthropogenic climate change. Very recently, datasets that blend together precipitation estimates from multiple sources such as gauge observations, satellite observations and even reanalyses have become available. Examples include MSWEP V2 (Beck et al., 2019), CHIRPS (Funk et al., 2015) and Shen et al. (2014). These datasets offer very high spatial and temporal resolution data with a reasonable long temporal record. However, these datasets may exhibit increased temporal variability due to the incorporation of various observational sources over time and do not include as many in situ station observations as the gauge-only datasets.

[revised manuscript text omitted]

2. If the coordinates were within 1 degree of each other and WMO IDs either matched or were missing and the correlation between the timeseries that overlap was greater than 0.99 and the overlapping timeseries themselves had at least 365 daily data records with a minimum of 10 days with precipitation greater than 1mm. A search radius of 1 degree was necessary to allow for many stations to be compared with each other in order to account for possible inaccuracies in station metadata (coordinates).

Note that the above algorithm can result in false matches as nearby stations can be highly correlated, however this will mainly be an issue in highly dense networks such as US. For the future version, a more quantitative measure of similarity

between station time series will be used. Also note that WMO station IDs do not change after a station is relocated to a site in the vicinity which can result in two stations in different locations merged together according to our criteria. On occasions where precipitation amount from a station was different between multiple sources, we prioritised data from higher quality sources and accepted values from these sources. The data qualities and hence priorities in descending order (highest quality first) are GPCC

5     data base, GPCC ASCII data files, Other data, GHCN-Daily data. This way if data from a higher quality source were missing, it was replaced with data from a matching station from a lower quality source but not vice versa. Note that this approach may induce inhomogeneities in the raw station data.

**2.4    Interpolation Method**

[revised manuscript text omitted]

20  outside the grid cell may be used for interpolation in some cases where density is low.

The 1950–2016 average Kriging error (KE) and coefficient of variation (CoV), and the data quality mask based on KE and CoV are shown for REGEN and REGEN40YR in figure 5. The CoV, defined as the ratio of the Yamamoto standard deviation and the precipitation estimate, is a normalised measure of the variance at each grid cell. The Kriging error is largest in regions with a low station density such as Greenland, Africa and South America and is larger for REGEN40YR compared to REGEN

25  as expected (figures 5a and 5b). Coefficient of variation, however, is comparable between REGEN and REGEN40YR. The largest CoV values (maximum of 2.33) are once again seen in Africa, South America, Greenland and Southeast Asia (figures 5c and 5d). This means that the variance between the grid cell estimate and the observations used for interpolation is more than twice as large as the average precipitation for these grids. Grids with CoV greater than 1.9 make up less than 0.05% (22 all together) of the grid cells with the mode of CoV being around 1.25. 
[revised manuscript text omitted]
 over-estimating compared to APHRODITE on November $1^{st}$ and November $9^{th}$. On the other hand, REGEN40YR captures more of the variability of SA-Obs compared to REGEN, especially November $3^{rd}$ onward. Interestingly, the spike on October $27^{th}$ is present in APHRODITE, REGEN and REGEN40YR but not in SA-Obs and the spike on November $8^{th}$ is present in SA-Obs, REGEN and REGEN40YR but not in APHRODITE. Finally based on a comparison of daily rainfall rates during tropical storm Amelia that made landfall in southern United States, there is also good agreement between REGEN, REGEN40YR and CPC CONUS (Chen and Xie, 2008; Xie et al., 2007; Chen et al., 2008) (figure 7d).

As a more detailed comparison, we calculated the difference in daily estimates between REGEN and the five regional datasets mentioned above (CPC CONUS, E-Obs V16, AWAP, APHRODITE, and SA-Obs). The five regional datasets were all regridded to the same 1 degree grid as REGEN and daily differences were calculated for each corresponding grid over the respective temporal periods of each regional dataset (CPC CONUS - 1950–2006, E-Obs V16 - 1950–2016, AWAP - 1950–2015, APHRODITE - 1950–2007, SA-Obs V1 - 1981–2014). Temporal correlations between REGEN and the respective regional

| Regional Dataset Name | Regional Dataset Stations | REGEN Stations in Region | REGEN40YR (Long Term) Stations in Region |
|---|---|---|---|
| APHRODITE | Daily Max 8000+ | 8551 (daily max 4985) | 1539 (daily max 1743) |
| SA-Obs V1.0 | 7956 | 2527 | 64 |
| E-Obs V16 | 17,468 | 28,338 | 11,261 |
| CPC CONUS | ~28,500 | 42,229 | 3940 |
| AWAP | Daily Max ~7500 | 12,993 (daily max 7509) | daily max of 3909 |

**Table 1.** Total number of stations interpolated by five regional datasets and the corresponding number of stations in each region interpolated by REGEN and REGEN40YR datasets over the entire time period of each respective regional datasets (CPC CONUS - 1950–2006, E-Obs V16 - 1950–2016, AWAP - 1950–2015, APHRODITE - 1950–2007, SA-Obs V1 - 1981–2014).

datasets were also calculated at each grid. The mean difference between REGEN and CPC CONUS is positive in eastern United States and negative in the west (figure 8a), the standard deviation (SD) of the daily difference is high in coastal areas (figure 8b), and the temporal correlation is high everywhere (figure 8c). The mean difference between REGEN and E-Obs V16 is positive in most regions across the E-Obs domain (figure 8d), the SD of the difference is higher in the South and in Iceland compared to the northern parts of the E-Obs domain (figure 8e), and the temporal correlations are higher in regions of high station density (such as central Europe, UK and Scandinavia) compared to low station density regions (such as Northern Africa and Turkey) (figure 8f). The mean difference between REGEN and AWAP is positive in northern and central Australia and negative elsewhere (figure 8g), the SD of the difference is high in the northern and eastern coastal areas of Australia (figure 8h), and the temporal correlation is high everywhere except for the low station density regions of central Australia (figure 8i). Note that similar to figure 7b the AWAP daily data had to be shifted a day backward once again for a more suitable comparison. The mean of the daily difference between REGEN and APHRODITE is positive in most regions, and both the mean and SD of difference showing higher values on the west coast of the Indian peninsula, the maritime continent and the high elevation Himalayan regions (figure 8j,8k). The temporal correlation between REGEN and APHRODITE is high in continental Asia and low in the maritime continent (figure 8l). Finally, the mean difference between REGEN and SA-Obs is positive in most regions of the SA-Obs domain with larger values of both the mean and SD of the difference in the maritime continent. Conversely, the temporal correlation between REGEN and SA-Obs is high in Northern Australia and low in the maritime continent. High differences between REGEN and all regional datasets are observed in coastal areas. Note that it is possible that this is an artefact of the regridding of the regional datasets to a 1 degree resolution.

A comparison of the number of stations interpolated by each of the five regional datasets mentioned above (APHRODITE, SA-Obs V1, E-Obs V16, CPC CONUS and AWAP), and the corresponding stations interpolated by REGEN and REGEN40YR in the respective regions of each datasets is shown in 1. In some cases, due a lack of available information, the daily maximum number of stations has been listed as opposed to the total number of stations for the entire time periods. In these cases, we also

provide the daily maximum number of stations interpolated by REGEN and REGEN40YR. REGEN and REGEN40YR interpolate less stations compared to APHRODITE and SA-Obs. This is particularly striking in Southeast Asia where REGEN40YR which interpolates only 64 stations compared to 7956 interpolated by SA-Obs. On the other hand REGEN interpolates more stations compared to E-Obs and CPC CONUS. Note that some of these stations, especially in the US, may be duplicates missed by our merging algorithm. Finally, there is little difference in the station network interpolated by REGEN and AWAP.

**3.3 Case study over Sub-Saharan Africa**

Based on the maps of Kriging error (figures 5a and 5b) the most data sparse regions of REGEN are Africa, South America, Greenland and northern Russia. Despite the sparsity of data, REGEN can still be useful to get estimates of daily rainfall in some parts of these regions. We use the country of Benin in sub-Saharan Africa as an example. Benin has a tropical climate receiving the majority of rainfall around the summer months of June-August (JJA). In the summer of 2008 Benin experienced catastrophic flooding events displacing around 150,000 people (WHO, 2010). The flooding started with heavy rainfall in the last week of July (IRIN, 2008). The timeseries of daily rainfall from 1950 to 2013 highlights 2008 as a year with the third highest rainfall on record based on REGEN (figure 9a) with the highest being in 1957. On comparison of the daily rainfall timeseries between 1957 and 2008 (figure 9b), the anomalous rainfall in late June and late July is apparent, even compared to 1957. This highlights REGEN's effectiveness in capturing the daily rainfall even in some parts of Sub-Saharan Africa. Note that the region of Benin is of higher quality compared to surrounding regions as it is not masked in the data quality mask (figure 5e).

**3.4 Comparison with existing global datasets of daily precipitation**

Finally, in this section the only other existing global gridded gauge-based datasets of daily precipitation are compared. The temporally averaged annual total, annual maximum precipitation, trends in annual total and trends in annual maxima are compared against NOAA Climate Prediction Center's (CPC) Unified Gauge-Based Analysis of Daily Precipitation (CPC-Global) (Chen and Xie, 2008; Xie et al., 2007; Chen et al., 2008) and GPCC Full Data Daily V1 (GPCC-FDD1) (figure 10). For comparability CPC-Global whose native resolution is 0.5 degrees was regridded to 1 degree to match the GPCC-FDD1 and REGEN. The temporal coverage of CPC-Global and GPCC-FDD1 is 1979–2017 and 1988–2013 respectively. The temporal averaging and comparison was therefore done over 1988–2013 which is the longest common period between the three datasets. As expected REGEN is more similar to GPCC-FDD1 and REGEN40YR compared to CPC-Global for both the means and trends of both indices. This is because REGEN and GPCC-FDD1 use the same interpolation method and for the most part even the same underlying data. The largest differences between the three datasets arise in data sparse regions in the high latitudes, Africa, South East Asia, and the high altitude regions in western South America. The spatial variability of the differences in annual total and annual maxima trends is higher compared to the spatial variability of differences in averages of the annual totals and annual maxima. Due to the lack of long term stations in Saharan Africa, differences in all four indices between REGEN and the long term station based REGEN40YR are larger compared to differences between REGEN and GPCC-FDD1 in northern Africa. Herold et al. (2016) showed CPC-Global produces lower annual totals compared to an ensemble of

observational datasets including GPCC-FDD1, satellite products and reanalyses. This is consistent with our results since the difference in annual totals between REGEN and CPC-Global are positive in majority of global land areas with the exception of northern North America and northern Africa.

Temporal and spatial correlation between REGEN and GPCC-FDD1 (figures 12a and 12b) are also higher compared to temporal and spatial correlation between REGEN and CPC-Global (figures 12c and 12d). Correlations between REGEN and CPC may be lower in parts where the underlying stations were shifted a day backward (see Appendix). Indeed, based on correlations between REGEN lagged +1 days and CPC (A1), the correlations are higher compared to 12d in and around the countries where data was shifted a day back (eg. Vietnam, Brazil, Uruguay, Peru, Suriname, Netherlands, Norway, Ukraine and Turkey). Correlations do not change compared to 12d in all regions where REGEN raw station data are not shifted. 
[revised manuscript text omitted]

Rainfall is highly variable and a 1 degree spatial resolution (roughly 10,000 sq km) dataset such as REGEN is unlikely to contain the information necessary for many typical local-to-regional rainfall applications. However, we note the actual rainfall
35   amounts in gridded datasets are subject to large uncertainties anyway (Herold et al., 2016), whereas estimates of variability

are more robust. We therefore believe REGEN will prove itself valuable for climatological applications including studies of climate variability and long-term changes in daily precipitation intensity and extremes, as it provides long temporal coverage of quasi-global daily precipitation observations. The biggest strength of REGEN is the long temporal coverage of quasi-global daily precipitation observations. Regional datasets are often developed by national meteorological organisations and often have

5  access to significantly more data than shared with Global archives such as GHCN-Daily and GPCC. For example the Spanish Meteorological Agency (AEMET) itself manages roughly over 9000 stations (Herrera et al., 2012) which is almost the same number of stations as those used by E-Obs for the entirety of Europe (around 12,000 guages at its maximum). Furthermore, Herrera et al. (2012) only used the high quality stations which accounted for roughly 30% of total stations available from the Spanish Meteteorological Agency (AEMET). Often the respective meteorological organisation also have the resources to more

10  thoroughly and in some cases even manually quality control the raw data. As a result, regional datasets (where available) may provide more accurate precipitation estimates than REGEN.

At the moment REGEN is not an operational product, meaning the analysis for REGEN was done as a single instance and there are currently no plans to update it regularly, such as on an annual or biennial basis.

Figure 13 reflects REGEN's strengths by showing annual totals and maxima and trends over the high quality regions over

15  the entire 63 year record of REGEN. Both the total annual precipitation and annual maxima based on REGEN are reasonable with higher totals and maxima in the known wet regions such as the tropics and lower totals and maxima in the known dry regions such as Saharan Africa (figures 13a and 13b). Trends in total precipitation based on REGEN (figure 13c) are also comparable to the trends in total precipitation shown in the IPCC's 5[th] Assessment report (figure 2.29, Hartmann et al. (2013)). The total annual precipitation, annual maxima and respective trends in the two indices based on the long term REGEN data

20  (REGEN40YR) (figures 13e, 13f, 13g and 13h) are also very similar to REGEN which suggests that the effects of station variations appear negligible at this scale (for trends and averages over 1950–2013) for the high quality grids. The trend maps shown in figure 13 have been masked based on the quality masks as shown in figures 5e and 5f.

REGEN provides precipitation estimates comparable to those from the currently most reliable datasets such as GPCC-FDD1. With a temporal coverage 152% longer than that of GPCC-FDD1's and a similar global land coverage, REGEN is highly

25  suitable for analysing climate change. We recognise that observations are not the "truth" but rather just our best estimates of it. REGEN and its variant REGEN40YR (which minimises station network variability) are therefore accompanied by various uncertainty estimates as well as a quality mask, allowing users a firm handle of the observational uncertainties in their analysis.

*Data availability.* REGEN AllStns V1-2019 (REGEN) and REGEN LongTermStns V1-2019 (REGEN40YR) data has now been published with unique Digital Object Identifiers (DOIs) https://dx.doi.org/10.25914/5ca4c380b0d44 & https://dx.doi.org/10.25914/5ca4c2c6527d2 re-

30  spectfully. Older versions of both datasets, REGEN AllStns V1.0 and REGEN LongTermStns V1.0, are also available (https://dx.doi.org/10. 25914/5b9fa55a8298c and https://dx.doi.org/10.25914/5b9fa67fce5d6 respectively), however, we recommend users use the newer versions. Both datasets can be acquired in netcdf format along with netcdfs of the quality masks via the Research Data Australia (RDA) web pages https://researchdata.ands.org.au/rainfall-estimates-gridded-v1-2019/1408744 and https://researchdata.ands.org.au/rainfall-estimates-gridded-v1-2019/

1408742 respectively. The RDA records contain further information about the datasets such as the dataset abstract, citation information, related organisations, grants, researchers and dataset managers (SC).

*Dataset License and Rights*. Non-Comercial License: CC-BY-NC-SA

Creative Commons - Attribution - Non Commercial - No Derivatives 4.0 International

5  http://creativecommons.org/licenses/by-nc-sa/4.0/legalcode

Access to this dataset is free, the users are free to download this dataset and share it with others and adapt it as long as they credit the dataset owners, provide a link to the license, and if changes were made, indicate it clearly and distribute their contributions under the same license as the original, commercial use is not permitted.

**Appendix A: List of countries for which the timestamps have been shifted a day back**

10  The countries for which the data are shifted a day back (e.g. data from $2^{nd}$ Jan are saved as $1^{st}$ Jan) are Angola, Antarctica, Argentina, Australia, Azerbaijan, Bahamas, Bangladesh, Barbados, Benin, Bolivia, Botswana, Brazil, Bulgaria, Burkina Faso, Cameroon, Chad, Chile, Costa Rica, Croatia, Denmark/Greenland, Ethiopia, French Polynesia, Gabon, Georgia, Indonesia, Islands in the Indian Ocean (IOT), Ivory Coast, Japan, Kenya, Libya, Madagascar, Malawi, Mali, Marshall Islands, Mauritania, Mozambique, Netherlands, Niger, Norway, Peru, Senegal, Slovenia, Solomon Islands, Sudan, Suriname, Tanzania, Tunisia, 15  Turkey, Ukraine, Uruguay, Vanuatu, Vietnam, Zambia, Zimbabwe.

[revised manuscript text omitted]

van den Besselaar, E. J. M., van der Schrier, G., Cornes, R. C., Iqbal, A. S., and Klein Tank, A. M. G.: SA-OBS: A Daily Gridded Surface Temperature and Precipitation Dataset for Southeast Asia, Journal of Climate, 30, 5151–5165, https://doi.org/10.1175/JCLI-D-16-0575.1, https://doi.org/10.1175/JCLI-D-16-0575.1, 2017.

Villarini, G. and Krajewski, W. F.: Sensitivity Studies of the Models of Radar-Rainfall Uncertainties, Journal of Applied Meteorology and Climatology, 49, 288–309, https://doi.org/10.1175/2009JAMC2188.1, 2010.

Westra, S., Alexander, L. V., and Zwiers, F. W.: Global Increasing Trends in Annual Maximum Daily Precipitation, Journal of Climate, 26, 3904–3918, https://doi.org/10.1175/JCLI-D-12-00502.1, http://journals.ametsoc.org/doi/abs/10.1175/JCLI-D-12-00502.1http://dx.doi.org/10.1175/JCLI-D-12-00502.1, 2013.

WHO: WHO | Floods in West Africa raise major health risks, http://www.who.int/mediacentre/news/releases/2008/pr28/en/#.WweHryUc8nM.mendeley, 2010.

Xie, P., Chen, M., Yang, S., Yatagai, A., Hayasaka, T., Fukushima, Y., and Liu, C.: A Gauge-Based Analysis of Daily Precipitation over East Asia, Journal of Hydrometeorology, 8, 607–626, https://doi.org/10.1175/JHM583.1, 2007.

Yamamoto, J. K.: An Alternative Measure of the Reliability of Ordinary Kriging Estimates, Mathematical Geology, 32, 489–509, https://doi.org/10.1023/A:1007577916868, 2000.

Yatagai, A., Xie, P., Alpert, P., Yatagai, A., Xie, P., and Development, P. A.: Development of a daily gridded precipitation data set for the Middle East, Advances in Geosciences, 12, 165–170, https://hal.archives-ouvertes.fr/hal-00297073, 2008.

Yatagai, A., Kamiguchi, K., Arakawa, O., Hamada, A., Yasutomi, N., and Kitoh, A.: APHRODITE: Constructing a Long-Term Daily Gridded Precipitation Dataset for Asia Based on a Dense Network of Rain Gauges, Bulletin of the American Meteorological Society, 93, 1401–1415, https://doi.org/10.1175/BAMS-D-11-00122.1, http://journals.ametsoc.org/doi/abs/10.1175/BAMS-D-11-00122.1, 2012.

Ziese, M., Rauthe-Schöch, A., Becker, A., Finger, P., Meyer-Christoffer, A., and Schneider, U.: GPCC Full Data Daily Version.2018 at 1.0°: Daily Land-Surface Precipitation from Rain-Gauges built on GTS-based and Historic Data, https://doi.org/10.5676/DWD_GPCC/FD_D_V2018_100, ftp://ftp.dwd.de/pub/data/gpcc/html/fulldata-daily_v2018_doi_download.html, 2018.

[Figure]

**Figure 3.** Grids showing percentage of days with at least 1 station in each decade for (figure 3a) REGEN, (figure 3b) REGEN40YR and (figure 3c) GPCC-FDD1. Gray areas indicate grids where no stations are present.

[Figure]

(a)

(b)

(c)

**Figure 4.** Percentage of records that (figure 4a) failed one or more quality control tests and were flagged and (figure 4b) were not used as input for interpolation due to missing monthly totals and hence missing anomaly values. Figure 4c shows a map of regions as used for figures 4b and 1a.

[Figure]

**Figure 5.** Kriging error (KE)(figures 5a and 5b), Coefficient of variation (CoV) (figures 5c and 5d) defined by the ratio of the Yamamoto standard deviation (Yamamoto, 2000) averaged over 1950-2016 and the daily precipitation averaged over 1950-2016, and masks based on the KE and CoV (figures 5e and 5f) based on REGEN (left Column) and REGEN40YR (right column) data.

[Figure]

**Figure 6.** Comparison of annual precipitation anomaly timeseries with monthly datasets. Anomalies were calculated relative to the average of daily precipitation totals over the entire time period (1950-2016) for each dataset.

[Figure]

**Figure 7.** Daily timeseries avearged over spatial regions of significant rainfall events. (figure 7a) Timeseries of daily rainfall during the great flood of 1968 over Southeast England. (figure 7b) Timeseries of daily rainfall during Cyclone Yasi in northeast Australia in 2011. (figure 7c) Timeseries of daily rainfall during typhoon Thelma in Philippines in 1991. (figure 7d) Timeseries of daily rainfall during tropical storm Amelia in US in 1978.

[Figure]

**Figure 8.** Mean (first column; a, d, g, j, m) and standard deviation (second column; b, e, h, k, n) of the difference in daily values (mm/day), and temporal correlation (third column, c, f, i, l, o) between REGEN and CPC CONUS (first row), REGEN and E-Obs V16 (second row), REGEN and AWAP (third row), REGEN and APHRODITE (fourth row), and REGEN and SA-Obs V1 (fifth row) over the respective periods of each regional dataset.

[Figure]

**Figure 9.** Timeseries of daily precipitation from REGEN averaged over Benin in Western Africa. Figure 9a shows the entire timeseries from 1950 to 2016 with the years containing the days with the highest three daily rainfall rates (1957, 1963 and 2008) shown in a darker shade. Figure 9b shows a comparison of the timeseries of daily rainfall between 1957 (year containing the day with the record highest rainfall based on REGEN) and 2008 (year during which the 2008 Benin floods occurred). Benin was chosen because of its good coverage of stations.

[Figure]

**Figure 10.** Percentage difference in averaged total annual precipitation (first column; figures 10c, 10e and 10h), averaged maximum annual precipitation (second column; figures 10d, 10f and 10g) between REGEN and GPCC (second row), REGEN and CPC (third row) and REGEN and REGEN40YR (fourth row) data. The first row shows the absolute values of total annual precipitation (figure 10a) and annual maxima (figure 10b) averaged over 1988 - 2013 (the longest common time period between the three datasets).

[Figure]

**Figure 11.** Percentage difference in total annual precipitation trends (first column; figures 11c, 11e and 11g), and annual maximum precipitation trends (second column; figure 11d, 11f and 11h) between REGEN and GPCC (second row), REGEN and CPC (third row) and REGEN and REGEN40YR (fourth row) data. The first column shows the absolute values of total annual precipitation trends (figure 11a) and annual maximum precipitation trends (figure 11b) averaged over 1988 - 2013 (the longest common time period between the three datasets).

[Figure]

**Figure 12.** Spatial (field) correlation at each daily time-step (first column; figures 12a, 12c and 12e) and temporal correlation between timeseries at each grid cell (second column; figures 12a, 12c and 12e) between REGEN and GPCC (first row), REGEN and CPC (second row) and REGEN and REGEN40YR Long term (third row) data. Comparisons are over the entire common temporal period between each dataset pair (1988–2013 for REGEN vs GPCC-FDD1, 1979–2016 for REGEN vs CPC, and 1950–2016 for REGEN vs REGEN40YR).

[Figure]

**Total annual precipitation**

**Annual maximum precipitation**

REGEN Average

(a)

(b)

REGEN Trend

(c)

(d)

REGEN40YR Average

(e)

(f)

REGEN40YR Trend

(g)

(h)

**Figure 13.** Total annual precipitation (figures 13a and 13e), maximum annual precipitation (figures 13b and 13f) and respective trends (PRCPtot; figures 13c and 13g and RX1DAY; figures 13d and 13h) averaged over 1950 to 2013 based on REGEN data (figures 13a, 13b, 13c and 13d) and REGEN40YR data that only interpolates stations with at least forty complete years of data (figures 13e, 13f, 13g and 13h).

[Figure]

**Figure A1.** Temporal correlations between REGEN and CPC, similar to figure 12d, but this time with CPC shifted a day backwards.